# Cluster wakes impact on a far distant offshore wind farm's power

Jörge Schneemann[1], Andreas Rott[1], Martin Dörenkämper[2], Gerald Steinfeld[1], and Martin Kühn[1]

[1]ForWind, Institute of Physics, Carl von Ossietzky University Oldenburg, Küpkersweg 70, 26129 Oldenburg, Germany
[2]Fraunhofer Institute for Wind Energy Systems, Küpkersweg 70, 26129 Oldenburg, Germany

**Correspondence:** Jörge Schneemann (j.schneemann@uol.de)

**Abstract.** Our aim with this paper was the analysis of the influence of offshore cluster wakes on the power of a far distant wind farm. We measured cluster wakes with long range Doppler light detection and ranging (lidar) and satellite synthetic aperture radar (SAR) in different atmospheric stabilities and analysed their impact on the 400 MW offshore wind farm «Global Tech I» in the German North Sea using supervisory control and data acquisition (SCADA) power data. Our results showed clear wind
speed deficits that can be related to the wakes of wind farm clusters up to 55 km upstream in stable and weakly unstable stratified boundary layers resulting in a clear reduction in power production. We discussed the influence of cluster wakes on the power production of a far distant wind farm, cluster wake characteristics and methods for cluster wake monitoring. In conclusion, we proved the existence of wake shadowing effects with resulting power losses up to 55 km downstream and encouraged further investigations on far reaching wake shadowing effects for optimized areal planning and reduced uncertainties
in offshore wind power resource assessment.

## 1 Introduction

Wind energy utilization at sea is an increasingly important part for the transition of the mainly fossil-based energy system towards renewable electricity generation. By the end of 2018 offshore wind turbines with a capacity of 6,382 MW were installed in German waters, 21,750 MW worldwide. A massive expansion of offshore wind energy utilization is expected in
many countries. Germany alone aims at an installed capacity of 15 GW by the year 2030 (Mackensen, 2019). Most of this capacity will be installed in the North and Baltic Sea mainly in large wind farm clusters. A wind farm cluster typically consists of several wind farms in direct vicinity, often operated by different parties, featuring different wind turbine types and different geometries. Here, we call a large accumulation of more than a hundred wind turbines a cluster.

Wind turbines extract energy from the atmosphere forming regions of reduced wind speed, so called wakes, behind them.
Wakes of single wind turbines merge to a wind farm or cluster wake (e.g. Nygaard, 2014). We use the term cluster wake for the merged wakes of a large number of wind turbines of either the same or different type with no individual wind turbine wake identifiable anymore. Downstream turbines within a wind farm (e.g. Barthelmie and Jensen, 2010) and in neighbouring downstream clusters (e.g. Nygaard and Hansen, 2016) experience reduced wind speeds and reduced power generation caused by wake shadowing effects. With a rising offshore wind energy utilisation cluster wake shadowing effects will occur to an
increasing degree, leading to power losses and uncertainties in offshore wind resource assessment.

Wind turbine wakes were subject of intensive research in the last decade. Wake measurements were mainly performed using the

remote sensing technique Doppler lidar (e.g. Aitken et al., 2014; Trabucchi et al., 2017; Bodini et al., 2017; Fuertes et al., 2018; Beck and Kühn, 2019), power analysis on the basis of SCADA data (e.g. Barthelmie and Jensen, 2010) or Doppler radar (e.g. Hirth et al., 2014). Furthermore, several numerical studies investigated wind turbine wakes using large eddy simulations (LES) (e.g. Churchfield et al., 2012; Abkar and Porté-Agel, 2015; Dörenkämper et al., 2015b; Lignarolo et al., 2016; Vollmer et al., 2016). In an unstable atmosphere e.g. in cold air over warm water, vertical turbulence leads to a well mixed boundary layer and causes a faster wake recovery. In stable conditions e.g. in warm air over cold water, wake deficits can last far downstream. Hansen et al. (2011), Dörenkämper et al. (2015b) and Lee et al. (2018) investigated wake recovery with respect to atmospheric stability and found an increased length of wakes in stable stratification. Optimized wind farm layouts on the basis of the prevailing wind rose and stability distribution to reduce wake effects are commonly used (e.g. Emeis, 2009; Turner et al., 2014; Schmidt and Stoevesandt, 2015).

Cluster wakes are recently coming into the scientific focus with an increased offshore wind energy utilisation. Due to the large dimensions of cluster wakes experimental investigations have been made with measurement systems capable to cover large areas like satellite synthetic aperture radar (SAR) (e.g. Hasager et al., 2015), research aircrafts (e.g. Platis et al., 2018) and Doppler radar (e.g. Nygaard and Newcombe, 2018). Numerical studies were carried out by implementing wind farms in mesoscale models (e.g. Fitch et al., 2012; Volker et al., 2015). Wakes of large offshore wind farm clusters over distances of more than 10 km were first observed using data from satellite SAR (Christiansen and Hasager, 2005). Li and Lehner (2013) and Hasager et al. (2015) analysed offshore wind farm wakes using SAR images and compared the long visible wakes to results of mesoscale models. Nygaard and Hansen (2016) analysed the power production of an offshore wind farm before and after the commissioning of a wind farm located 3 km to the west on the basis of SCADA data and discovered power losses caused by wakes of the upstream wind farm in the first rows of the downstream wind farm. Nygaard and Newcombe (2018) used dual-Doppler wind radar to measure the inflow and the wake of an offshore wind farm and found wind speed deficits up to the maximal achievable downstream distance of 17 km possible with the used setup. They analysed a case with steady wind direction and speed and observed the cluster wake for over one hour, stability information was not available. Platis et al. (2018) used *in situ* measurements taken with a research aircraft on hub height behind offshore wind farm clusters in the German North Sea and identified wakes with lengths of up to 55 km under stable atmospheric conditions, up to 35 km in neutral conditions and up to 10 km in unstable conditions. Siedersleben et al. (2018b) used the same flight measurements as Platis et al. to evaluate a wind farm parametrization (Fitch et al., 2012) in the numerical Weather Research and Forecasting model (WRF) that is well established in wind energy applications (e.g. Pryor et al., 2018b; Witha et al., 2019; Dörenkämper et al., 2015a). Additionally they presented an analysis of aircraft wake measurements in five different heights 5 km downwind of the cluster. The wake deficit existed in all considered height levels also 50 m above the upper tip height of the rotor. Siedersleben et al. (2018a) investigated the micrometeorological consequences of cluster wakes due to mixing effects in the atmosphere using the flight measurements from Platis et al.. Pryor et al. (2018a) evaluated the downstream impact of large onshore wind farms in North America using the wind farm parametrization by Fitch et al. in convection permitting mesoscale WRF simulations. Lundquist et al. (2019) analysed the physical, economic and legal consequences of wake effects between large onshore wind farms with sizes of more than a hundred megawatt each.

Wind farm cluster wakes in the far field of more than 20 km downstream have not been measured over longer periods. Satellite SAR just offers the possibility to take snap shots of the wind field. Doppler radar has been deployed on the coast monitoring a near shore wind farm (Nygaard and Newcombe, 2018) but not in an offshore wind farm to use the full measurement range for wake analysis. Doppler lidar, which successfully monitored wind turbine wakes, was considered not to be able to achieve the measurement range needed to investigate full cluster wakes. Furthermore, the influence of cluster wakes on the power production of far downstream wind farms has not been analysed. The influence of atmospheric stability on the development and recovery of cluster wakes has not been studied in detail.

The objective of this paper is to analyse whether offshore cluster wakes have a significant and continuous influence on the power generation of a far downstream wind farm, and how this influence depends on atmospheric stability. For this purpose we investigated two exemplary cases of cluster wakes approaching the 400 MW wind farm «Global Tech I» in the North Sea during situations with different atmospheric stabilities by means of four synchronized data sets, namely

1. large-area satellite SAR wind data,

2. continuous platform-based long range Doppler lidar wind monitoring,

3. operational data of the wind farm «Global Tech I» and

4. meteorological measurements for atmospheric stability characterisation.

We follow Platis et al. (2018) in their definition of the cluster wake deficit as the difference of the wind speeds from the manually selected wake region and a neighbouring free flow region since the inflow wind speed of the wake generating cluster as reference is typically not known. Furthermore, regional and temporal differences in the wind field distort a comparison of the far distant points in front of and far behind a cluster. Therefore the adjacent regions in and aside the wakes are compared. Wake and free flow regions are identified manually in this analysis.

The paper is structured as follows. Section 2 introduces the experimental setup in the North Sea, measurements taken with Lidar, SAR and meteorological sensors as well as data processing. Section 3 presents two exemplary cluster wake cases affecting the wind farm «Global Tech I». In Section 4 we discuss the influence of cluster wakes on the power production of a far downstream wind farm as well as cluster wake characteristics and methods for cluster wake monitoring. Section 5 concludes on the findings and closes the paper.

## 2 Methods

In this study different data sources have been used: meteorological measurements, wind farm production data (supervisory control and data acquisition, SCADA) and remote sensing data from a Doppler lidar (light detection and ranging) measurement campaign and satellite SAR (synthetic aperture radar) data. A description of these data sources is given in this section. Our measurement campaign started in late July 2018 planned to last one year. Measurements we present in this paper were taken on 11 October 2018 and 6 February 2019. All measurement data in this study was recorded in UTC.

## 2.1 Wind farms and SCADA data

With status of early 2019 several offshore wind farms were installed mainly in clusters in the German and Dutch North Sea. Focus of this work is on the effects on the 400 MW wind farm «Global Tech I» (GT I), which is one of the world's most distant offshore wind farms with a coastal distance of more than 100 km. We analyse the impact of two large wind farm clusters, namely the 802 MW «BorWin» cluster located about 25 km southwest and the 914 MW «DolWin2» cluster 55 km southeast on the wind farm GT I.

Figure 1 gives an overview of the region around GT I while Figure 2 displays its layout. All maps we show in the following,

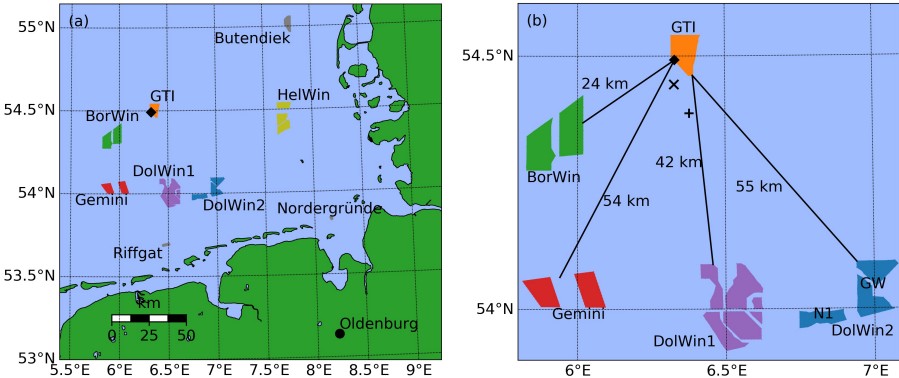

**Figure 1.** (a) Overview of the considered area in the southern North Sea with wind farms and clusters shown. (b) Close view on GT I and neighbouring wind farm clusters. The position of the lidar in GT I on turbine GT58 (filled ⋄) and the platforms OSS «Hohe See» (×) and «BorWin gamma» (+) are marked, distances to upstream clusters are shown. We measured wakes of all clusters in (b) and exemplary present the wakes of the «BorWin» and the «DolWin2» clusters in this work. Information on the wind farms and full names are listed in Table 1.

except of Figure 1, were transferred to the Gauss Krüger coordinate system and the origin was shifted to the lidar position at turbine GT58 in GT I (Figure 2). Table 1 summarizes the main characteristics of the wind farms and clusters in the region. In direct southwestern vicinity of GT I the associated wind farms «Hohe See» and «Albatros» were under construction during the period of our measurement campaign with several transition pieces and a substation but no wind turbine towers installed. The first turbine was erected 6 April 2019 (EnBW, 2019). The position of the «Hohe See» offshore sub station (OSS) is marked in the following plots (×). The installation of the 900 MW high voltage direct current (HVDC) platform «BorWin gamma» in the southeast corner of «Hohe See» was completed on 11 October 2018 (Petrofac, 2018), we mark its position (+).

For the wind farm GT I, ten minute averaged SCADA data was available during the period of the measurements. Data of turbines in normal operation was considered, turbines with curtailed power below rated power were excluded from the analysis based on a SCADA status flag, a curtailment signal and consideration of pitch angles. For the wind farms «BARD Offshore 1», «Gode Wind 1+2» and «Nordsee One» we obtained hourly production data from Fraunhofer ISE (2019) and checked the

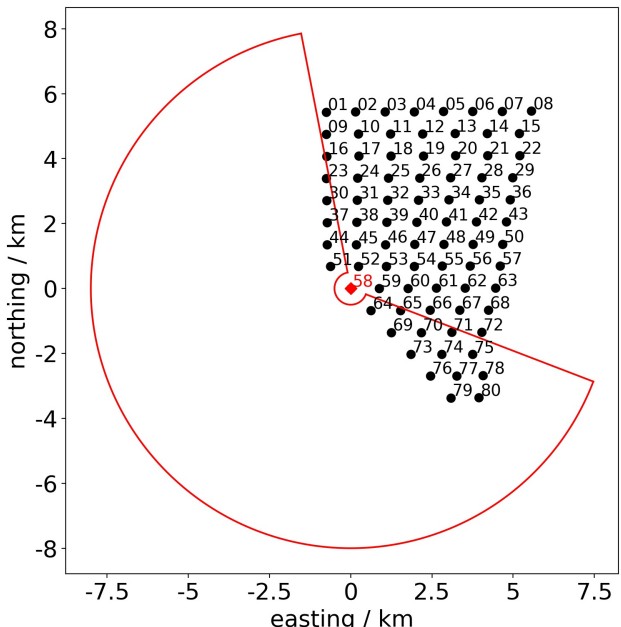

**Figure 2.** Layout of the wind farm «Global Tech I» with turbine numbers. The turbine GT58 where we positioned the lidar is marked in red (◇). The achievable sector for lidar measurements is drawn.

operational status. We analyse wind turbine power differences using the z-score

$$z_{P_i} = \frac{P_i - \overline{P_{\mathrm{up}}}}{\sigma_{P_{\mathrm{up}}}} \tag{1}$$

being the difference of the $i$th turbine's power $P_i$ and the mean power of the turbines in the first row facing the wind direction (upstream turbines) $\overline{P_{\mathrm{up}}}$ normalized with the standard deviation of the power of the upstream turbines $\sigma_{P_{\mathrm{up}}}$ within the consid-

5  ered time span. Advection through the farm is not considered. We use the upstream turbines to calculate the z-score instead of the turbines of the whole farm to avoid distortion by inner farm wake effects.

## 2.2 Lidar measurements

We used a scanning long range Doppler lidar system of type Leosphere Windcube200S (Serial no. WLS200S-024) in this study. The lidar system emits laser pulses into the atmosphere and analyses the light backscattered by aerosols for a Doppler

10  shift proportional to the radial wind velocity in beam direction $v_{\mathrm{r}}$. The lidar is able to process wind speed information in >200 different ranges on the beam called range gates. For each range gate the radial wind speed $v_{\mathrm{r}}$ and the carrier-to-noise ratio (CNR) as a measure of the signal quality are stored. The lidar's scanner is able to point the beam in any desired direction in the hemisphere above and partly below the device.

We installed the lidar system on the transition piece (TP) of wind turbine GT58 in GT I (filled ◇ in Figures 1 and 2). The

15  height of its scanner was approximately 24.6 m above mean sea level (MSL), 67.0 m below hub height and 9.0 m below lower

**Table 1.** Overview of offshore wind farms considered in this work (status June 2019). The wind farms «Borkum Riffgrund 2» (Orsted, 2018) and «Merkur Offshore» (Merkur Offshore, 2018) were in the commissioning phase and partly fed into the grid during our measurements, therefore they are marked with smaller symbols in the relevant plots in this paper. $D$: rotor diameter, $h_{\mathrm{H}}$: hub height, $P_{\mathrm{r}}$: rated power per turbine, No.: number of turbines per wind farm, $\Sigma P_{\mathrm{r}}$: rated power of wind farm. The numbers for the hub height are related to different reference levels, namely lowest astronomical tide (LAT), mean sea level (MSL) or just «over water». These differences are not further considered here since the difference between LAT and MSL is typically around 2 m in the North Sea.

| Name | Short | Turbine | $D$/ m | $h_{\mathrm{H}}$/m | $P_{\mathrm{r}}$/ MW | No. | $\Sigma P_{\mathrm{r}}$/ MW |
|---|---|---|---|---|---|---|---|
| Global Tech I | GTI | AD 5-116 | 116 | 92 | 5.0 | 80 | 400 |
| **BorWin Cluster (802 MW)** | | | | | | | |
| BARD Offshore 1 | BO1 | BARD 5.0 | 122 | 90 | 5.0 | 80 | 400 |
| Veja Mate | VM | SWT-6.0-154 | 154 | 103 | 6.0 | 67 | 402 |
| **Gemini Cluster (600 MW)** | | | | | | | |
| Buitengaats | BG | SWT-4.0-130 | 130 | 89 | 4.0 | 75 | 300 |
| Zee Energie | ZE | SWT-4.0-130 | 130 | 89 | 4.0 | 75 | 300 |
| **DolWin 1 Cluster (1,416 MW)** | | | | | | | |
| Trianel Windpak Borkum | TWB | AD 5-116 | 116 | 92 | 5.0 | 40 | 200 |
| alpha ventus | av | AD 5-116 | 116 | 90 | 5.0 | 6 | 30 |
| | | 5M | 126 | 92 | 5.0 | 6 | 30 |
| Borkum Riffgrund 1 | BR1 | SWT-4.0-120 | 120 | 87 | 4.0 | 78 | 312 |
| Borkum Riffgrund 2 | BR2 | V164-8.0 | 164 | 111 | 8.0 | 56 | 448 |
| Merkur Offshore | MO | Haliade 150-6 | 150 | 103 | 6.0 | 66 | 396 |
| **DolWin 2 Cluster (914 MW)** | | | | | | | |
| Nordsee One | N1 | 6.2M-126 | 126 | 90 | 6.15 | 54 | 332 |
| Gode Wind 1+2 | GW | SWT-6.0-154 | 154 | 110 | 6.0 | 97 | 582 |

blade tip height of the turbine. Figure 3 displays a picture of the lidar installed in GT I. The lidar performed horizontal plan position indicator (PPI) scans (elevation angle $\varphi$ was $0°$) with continuous scanner movement in different azimuthal sectors of $150°$ width upstream with two different settings A and B as listed in Table 2. We started with the slower scenario A aiming for a high measurement range. Later we optimized the measurements using scenario B being four times faster and achieved similar ranges. In both scenarios the laser beam is scanned over an angle of $2°$ per measurement leading to spatial averaging perpendicular to the line of sight direction. After performing a scan the lidar needs a few seconds to reset and start the next scan. Every few hours it performs a homing procedure of the scanner to assure precise orientation. The laser pulse length used in both scenarios was 400 ns leading to a probe volume of approximately 70 m in the beam direction. The range gate spacing is listed in Table 2. The offset in the azimuthal direction between geographic north and the lidar's north was corrected by scanning distant wind turbines in GT I with known positions («hard targeting»). The resulting error in the azimuthal orientation $\Delta\varphi$ was smaller than $0.1°$ and is therefore neglected.

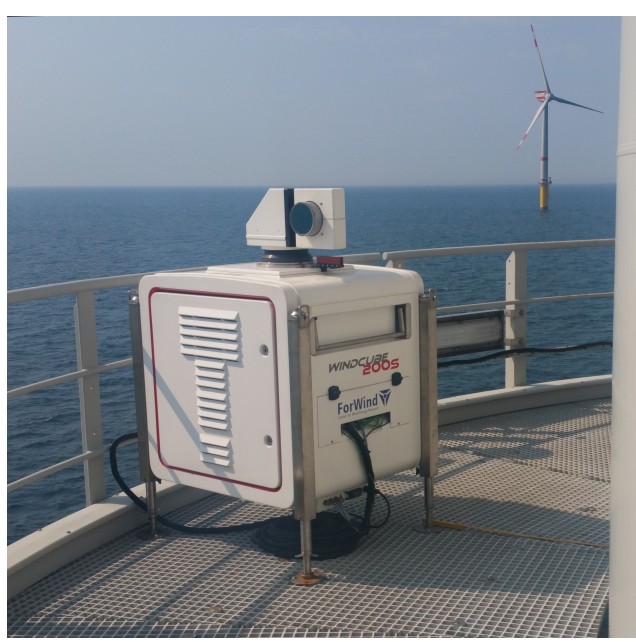

**Figure 3.** Lidar system Windcube200S on the transition piece of wind turbine GT58 in the offshore wind farm «Global Tech I». On the right side of the image the tower of the turbine is visible while turbine GT51 northwest of GT58 can be seen in the background (c.f. GT I layout in Figure 2). (Stephan Voß, ForWind)

**Table 2.** Overview of the different settings for the lidar PPI scans. Both scenarios covered different sectors of $150°$ width. Range gates are listed as minimal range : spacing : maximal range. Range gates are also referred to as "measurement points" in the following.

| Scenario | Pulse lengths / ns | Acquisition time / s | $\dot{\varphi}/°$/s | Scan duration / s | Range gates / m |
|:---:|:---:|:---:|:---:|:---:|:---:|
| **A** | 400 | 8.0 | 0.25 | 600 | 1000:50:12000 |
| **B** | 400 | 2.0 | 1.0 | 150 | 500:35:8000 |

The lidar was well aligned on the pitch and roll axis, errors were checked using the method of sea surface levelling (Rott et al., 2017). The resulting maximal error in the elevation $\Delta\vartheta$ was less than $0.1°$. An additional error in the elevation angle of the lidar measurement occurs from a small movement of the TP due to the thrust on the rotor with a maximum of $0.1°$.

When regarding the height of the measurement locations the curvature of the earth must be taken into account for the ranges
5    achieved. The error introduced raises quadratically with range and reaches $\Delta h_8 = 5.02$ m in a distance of 8 km and of $\Delta h_{10} = 7.85$ m in a distance of 10 km. The measurement errors we describe here can be neglected for the mainly qualitative analysis in this work.

## 2.3 Lidar data processing

Lidar scans were individually filtered on CNR minimal and maximal thresholds, a maximum range and a minimal data density in the $v_\mathrm{r}$-CNR-plane (similar to Beck and Kühn, 2017). For each PPI scan the mean wind direction was determined by fitting a cosine function to all radial speeds $v_\mathrm{r}$ of the scan over their azimuth angles $\varphi$. All $v_\mathrm{r}$ were then transformed back to the absolute
wind speed $v_\mathrm{a}$ in mean wind direction assuming the perpendicular wind component to vanish using

$$v_\mathrm{a} = v_\mathrm{r} / \cos(\varphi_\mathrm{diff}) \tag{2}$$

with $\varphi_\mathrm{diff}$ being the difference angle between the beam direction and the mean wind direction. Sectors with measurement ranges almost perpendicular to the wind direction ($|\varphi_\mathrm{diff}| > 75°$) were excluded from the analysis because of an increasing error due to an overestimation of flow components perpendicular to the wind direction. We plot single lidar scans on their original polar
grid. To obtain averaged lidar wind fields we transferred the $v_\mathrm{a}$-lidar data of each regarded scan to a Cartesian grid with a resolution of $50\,\mathrm{m} \times 50\,\mathrm{m}$ triangulating the data points and on each triangle performing linear barycentric interpolation to the grid points. We then calculated the cubic (or power) average on each grid point. Due to slightly changing wind directions in the averaging interval points at the border of the scans were just included in the further analysis if no scan (scenario A) or less than 10 scans (scenario B) did not contribute at the grid point.

## 2.4 Atmospheric stability and meteorological data

Meteorological measurements of atmospheric stability are uncommon in offshore wind farms. Different methods for the derivation of stability exist (c.f. Rodrigo et al. (2015) for an overview). We applied the *bulk Richardson method from profile measurements* according to Emeis (2018) based on the tropical observations of Grachev and Fairall (1997). We used the wind speed $v_\mathrm{TP}$ and the temperature $T_\mathrm{TP}$ on the height of the transition piece $z_\mathrm{TP}$ and the difference of the virtual potential temperatures
at the height of the TP and at sea level $\Delta\Theta_\mathrm{v} = \Theta_\mathrm{v,TP} - \Theta_\mathrm{v,SST}$ (c.f. Appendix A) to derive the dimensionless bulk Richardson number

$$\mathrm{Ri_b} = \frac{g}{\Theta_\mathrm{v,TP}} \frac{z_\mathrm{TP}\Delta\Theta_\mathrm{v}}{v_\mathrm{TP}^2} \tag{3}$$

where $g$ is the gravity acceleration. The dimensionless stability parameter

$$\zeta = \begin{cases} \dfrac{10\mathrm{Ri_b}}{1 - 5\mathrm{Ri_b}} & \mathrm{Ri_b} > 0 \\ 10\mathrm{Ri_b} & \mathrm{Ri_b} \leq 0 \end{cases} \tag{4}$$

and the stability classification in Table 3 were chosen for stability categorization.

  To be able to estimate $\zeta$ we operated sensors for air pressure (Vaisala PTB330) as well as temperature and relative humidity (Vaisala HMP155) on the TP of turbine GT58. In one case (c.f. Section 3.2.1) we used meteorological measurements from the nacelle of turbine GT58 provided by the wind farm operator as a second source of data to derive the stability parameter at height of the nacelle $\zeta_\mathrm{nac}$ using the same methodology as described above. A buoy for the measurement of the sea surface

**Table 3.** Classification of atmospheric stability as suggested by Sorbjan and Grachev (2010).

| Stability category | Range |
|---|---|
| very stable | $0.6 < \zeta < 2.0$ |
| stable | $0.2 < \zeta < 0.6$ |
| weakly stable | $0.02 < \zeta < 0.2$ |
| near neutral | $-0.02 < \zeta < 0.02$ |
| weakly unstable | $-0.2 < \zeta < -0.02$ |
| unstable | $-0.6 < \zeta < -0.2$ |
| very unstable | $-2.0 < \zeta < -0.6$ |

temperature $T_{\mathrm{SST}}$ was available from 9 August 2018 until 31 January 2019. We compared the measurements with the OSTIA data set (Donlon et al., 2012) both resampled to a 30 minute interval (mean values for the buoy data, linear interpolation for the daily available OSTIA data set) and found a mean difference of 0.19 K. Since the buoy was not available during the whole lidar measurement campaign, we use $T_{\mathrm{SST}}$ from the OSTIA data set to derive $\zeta$. The wind speed on the height of the TP $v_{\mathrm{TP}}$ for

5  the purpose of atmospheric stability analysis was calculated from horizontal lidar PPI scans as described in Section 2.3 using data with a measurement range smaller than 3000 m. These measurements took place within the approaching cluster wakes, when present. This influences the calculation of the stability parameter but we see the wake as part of the inflow and do not try to correct for it. We averaged meteorological measurements to 30 minute intervals.

For a comparison of the potential power $P_{\mathrm{pot}}$ in the wind with the power harvested by free flow turbines we had to transfer

**Table 4.** Overview of the available meteorological quantities to derive the stability parameter $\zeta$. Availabilities disregard shorter data gaps. If no end time is stated measurements are ongoing with date of 01 August 2019. Additional the data from mesoscale simulations similar to the NEWA data set were available but not listed in this table.

| quantity | symbol | sensor / source | height | availability period |
|---|---|---|---|---|
| air temperature | $T_{\mathrm{TP}}$ | HMP155 | $z_{\mathrm{TP}} = 24.6$ m MSL | 23.07.2018 - |
| air humidity | $\mathrm{rH}_{\mathrm{TP}}$ | HMP 155 | $z_{\mathrm{TP}} = 24.6$ m MSL | 23.07.2018 - |
| air pressure | $P_{\mathrm{TP}}$ | PTB 330 | $z_{\mathrm{TP}} = 24.6$ m MSL | 23.07.2018 - |
| wind speed | $v_{\mathrm{TP,lidar}}$ | Lidar PPI scans | $z_{\mathrm{TP}} = 24.6$ m MSL | 17.08.2018 - (dep. on scan scenario) |
| sea surface temperature | $T_{\mathrm{SST,buoy}}$ | buoy next to GT58 | sea surface | 09.08.2018 - 31.01.2019 |
| sea surface temperature | $T_{\mathrm{SST,OSTIA}}$ | OSTIA data set | sea surface | 2018 - 2019 |

10  wind speeds from measurement heights ($z_{\mathrm{SAR}} = 10$ m, $z_{\mathrm{TP}} = 24.6$ m) to hub height $z_{\mathrm{hub}} = 91.6$ m. Following Emeis (2018) we used the logarithmic wind profile

$$u(z) = \frac{u_*}{\kappa} \cdot \left( \ln \frac{z}{z_0} - \Psi_m(z/L) \right) \tag{5}$$

with a correction function $\Psi_m(z/L)$ to account for the atmospheric stability to calculate the vertical wind profile. We used mesoscale data with a setup very similar to the production runs of the «New European Wind Atlas» (NEWA, c.f. Witha et al. (2019) and NEWA (2019)) internally deriving the roughness length $z_0$ using Charnock's relation. We obtained the Obukhov length $L$ from the stability parameter $\zeta = z_{\mathrm{TP}}/L$. The von Karman constant reads $\kappa = 0.4$. The friction velocity $u_*$ was then

calculated for the given pair of wind speed and height, e.g. $z_{\mathrm{TP}}$ and $u_{\mathrm{TP}}$ from Equation 5. The wind speed on hub height was afterwards converted to the theoretical potential power $P_{\mathrm{pot}}$ using a power curve $P_{\mathrm{est}}(v) = c \cdot v^3$ with the constant $c$ derived from power data in the partial load range. We do not curtail $P_{\mathrm{pot}}$ at rated wind speeds allowing it to be larger than rated power.

## 2.5   SAR wind data

Satellite SAR remotely measures the roughness of the sea surface. Using a geophysical model to estimate wind direction,

wind speeds over the ocean can be derived. In this work, we use publicly available already processed wind data from the Copernicus SAR-satellite Sentinel-1A. The algorithm for wind field processing is described in Mouche (2011), an overview of its performance is given in ESA (2019) and the data product including quality flags is described in Vincent et al. (2019). Wind data at 10 m height is processed on a grid with a spatial resolution of $1{\times}1$ km$^2$. Wind speed estimates are in range from 0 m s$^{-1}$ to 25 m s$^{-1}$ with a root mean square error (RMSE) smaller than 2.0 m s$^{-1}$ and wind direction estimates have an

RMSE below 30°. The spatial coverage of the SAR images and the processed wind fields is 170 km$\times$80 km minimum with a revisit time in the order of days. A quality flag for the wind estimate (*owiWindQuality*, 0: high quality, 1: medium quality, 2: low quality, 3: bad quality, c.f. Vincent et al. (2019)) is provided within the data product. We use data with a quality flag $\leq 2$. For the calculation of the potential power on hub height (c.f. Section 2.4) we added constant wind speed values within the measurement accuracy to the SAR wind data to match the actual power production.

## 3   Results

In this section we present an analysis of wake situations of the «BorWin» cluster on 6 February 2019 and of the «DolWin2» cluster on 11 October 2018 based on Sentinel-1 SAR wind data, lidar measurements and SCADA power data of the wind farm GT I.

### 3.1   «BorWin» cluster wake on 6 February 2019

The «BorWin» cluster is located approximately 24 km upwind of GT I in southwesterly direction. We measured wakes from the cluster approaching GT I in stable stratified situations during our measurement campaign. Here we present a stably stratified situation in late winter 2018/2019 with low variation in the wind direction allowing us to analyse lidar scans of the same situation over a period of a couple of hours.

### 3.1.1 Meteorological conditions

In Figure 4 we plot the measured wind speed and direction, air pressure, temperature and humidity as well as the sea surface temperature from the OSTIA data set and the derived stability parameter $\zeta$ during 6 February 2019. On that day the frontal system of a cyclone southwest of Iceland crossed the German Bight. The warm front passed GT I in the morning bringing air temperatures of about 6.9 °C in the warm sector over the 6.1 °C cold sea stabilizing the boundary layer. With decreasing humidity and disappearing fog good lidar availability was achieved starting at approximately 10:00 (short humid/foggy period of bad measurements around 12:00) with clear wakes of the «BorWin» cluster visible in the lidar scans. In the afternoon we choose a period with relatively constant wind direction from 13:35 to 16:12 for analysing the averaged wake effects over a longer period of about 2.5 hours. The period with stable stratification ended with the passage of the cold front at approximately 17:15.

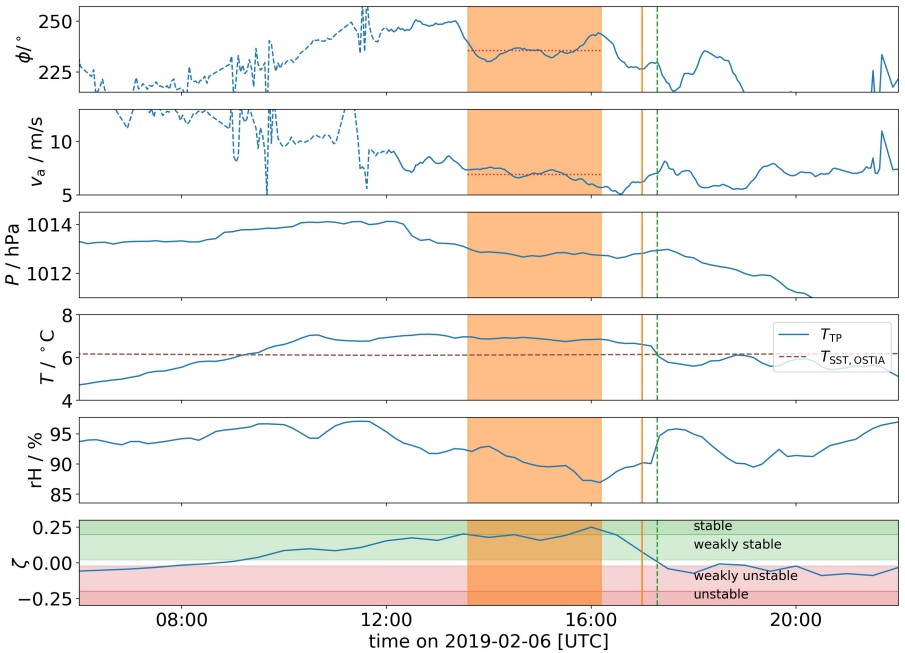

**Figure 4.** Meteorological data at the lidar location on the height of the TP (24.6 m MSL) of turbine GT58 on 6 February 2019. Top to bottom: wind direction $\phi_{\text{TP,lidar}}$, wind speed $v_{\text{TP,lidar}}$, air pressure $P_{\text{TP}}$, air and sea surface temperature $T_{\text{TP}}$ and $T_{\text{SST,OSTIA}}$, relative humidity $\text{rH}_{\text{TP}}$ and the dimensionless stability parameter $\zeta_{\text{TP}}$. Measurement times are marked as follows: vertical dashed line: SAR image (Figure 5), vertical solid line: single lidar scan (Figure 6), shaded interval: averaged lidar wind field (Figure 7). Mean wind speed and direction in the averaged lidar interval are marked by red horizontal dotted lines. Dashed lines in wind speed and direction indicate moist/foggy periods with reduced lidar data availability.

### 3.1.2 SAR wind data

Figure 5 displays the analysis of a wind field derived from the measurement of the Copernicus satellite Sentinel-1A, which passed the German Bight at the end of the stable stratified period on 6 February 2019 as an overview of the wind field in the region around GT I. The wake of the «BorWin» cluster is clearly visible and extends approximately 24 km downstream until it partially hits the wind farm GT I. Further downstream of GT I an even higher wake deficit of the merged wakes of the «BorWin» cluster and GT I can be observed. The virtual wake cut (Figure 5c) reveals a sharp transition from higher to lower wind speeds at the edge of the wake, a deficit in the SAR wind speed of $0.9 \mathrm{~m\,s^{-1}}$ is observed. Since the wake just partially hits GT I it separates the farm in two regions, one in free flow and one affected by the wake. The turbines in free flow in the northwestern and southern corner of GT I produce significantly more power ($> 2\sigma_P$) than the first upstream row turbines produce in average (Figure 5b). We confirm this result with the comparison of the 10 minute power of the upstream row turbines with the potential power on hub height derived from the inflow wind speed (Figure 5d) which agrees well. Within the wake affected region in GT I typical inner farm wake effects are visible through a power decrease in downstream direction (e.g. Barthelmie and Jensen, 2010, Figure 5b) which are different in the northern and southern part of the farm due to different turbine spacings in wind direction.

### 3.1.3 Lidar wind fields

In Figure 6 we present the analysis of a single lidar scan of the inflow of GT I. We observe a clear edge between high wind speeds in the undisturbed flow and lower wind speeds in the wake of the «BorWin» cluster causing a clear separation of power production in the wind farm GT I in a free flow and a wake region (Figure 6b). The virtual wake cut in Figure 6c illustrates the sharp transition region of just a few hundred meters width and highlights the wake deficit of $3.9 \mathrm{~m\,s^{-1}}$ or 40.5 %. The potential power on hub height derived from the inflow wind speed corresponds well with the power generated by the upstream row of turbines in the regarded ten minute interval (Figure 6d). The two northerly upstream turbines are in the region of free flow and produce with $> 2\sigma_P$ significantly more power than the turbines being influenced by the «BorWin» wake.

In Figure 7 we present an averaged lidar wind field calculated from 60 consecutive scans like the one in Figure 6 in a period of approximately 157 minutes with relatively constant wind direction (cf. shaded areas in Figure 4) to demonstrate the steadiness of the «BorWin» wake and its influence on power production. The wind speed along the virtual cut through the wind field in Figure 7c reveals a strong average wake deficit of $2.3 \mathrm{~m\,s^{-1}}$ equivalent to 24.7 %. The transition region from wake flow to free flow is about 3 km wide resulting from the small changes in wind direction and thus the slightly different positions of the wake during the averaging time. Aside the clear visible northerly edge of the «BorWin» wake the southerly edge can be observed in the southerly corner of the lidar wind field and correspondingly in the wake cut (Figure 7c). Wind speeds recover on both sides of the wake to similar values just above $9 \mathrm{~m\,s^{-1}}$. The average power of the GT I turbines reveals a clear reduction in the wake affected region (Figure 7b). The turbines in free flow produce ($> 2\sigma_P$) above the average. Comparing the potential power on hub height along the wake cut together with the average power of the upstream row turbines (Figure 7d) we find a slight overestimation of the potential power in the wake region and an overestimated increase of the turbine power in the

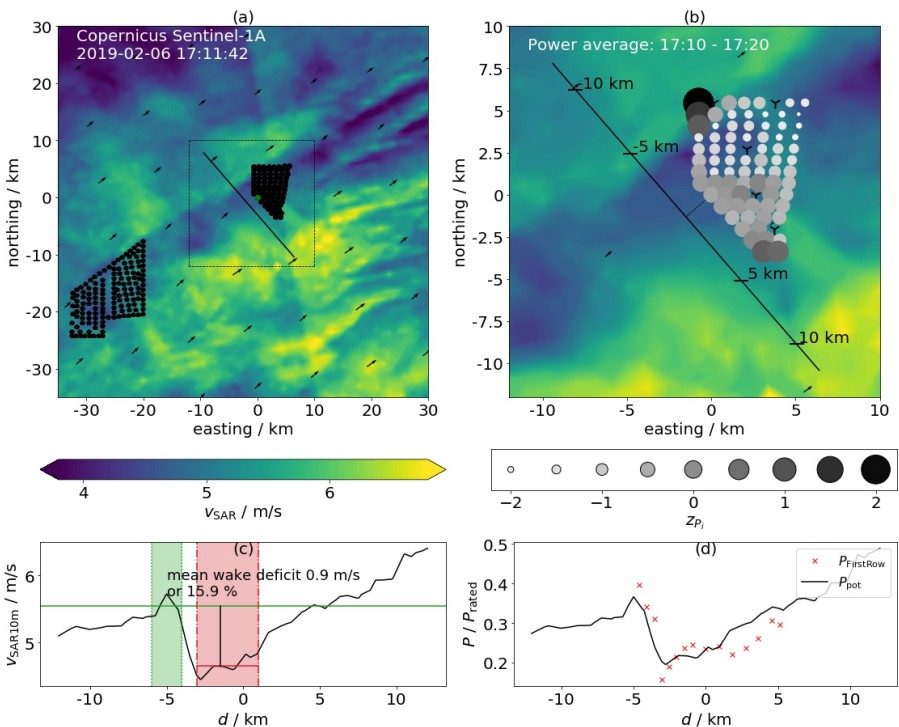

**Figure 5.** Sentinel-1A Ocean Wind Field (Copernicus Sentinel data [2019]), measurement taken 6 February 2019 17:11:42 UTC. (a) Overview of the «BorWin» cluster and «Global Tech I». (b) Close look on the «BorWin» wake hitting GT I. The solid line marks a virtual wake cut 2000 m upstream of turbine GT58 on which the wind field is evaluated. Marked distances correspond to the x-axis of sub figures c) and d). The z-score of the turbine power $z_{P_i}$ (cf. Equation 1) is shown in grey scales for the relevant ten minute period (17:10 - 17:20), markers scale with $z - z_{\min}$. Numbers of upstream turbines to calculate the z-score are 1, 9, 16, 23, 30, 37, 44, 51, 58, 64, 69, 73, 76, 79, 80. Turbines not operating the full period or operating at curtailed power are excluded and marked (Y-shaped marker). c) Wind speeds along the wake cut from b). Wake and free stream are shaded (regions selected manually). d) potential power on hub height along the wake cut (solid line) together with the power produced by the upstream turbines in GT I within the regarded ten minute interval with turbine positions projected to the wake cut. A constant value of $1.0 \, \mathrm{m\,s^{-1}}$ was added to $v_{\mathrm{SAR},10m}$ for the calculation.

transition region. The position of the transition onset in the estimated power from the wind field and the measured power from the turbines agree well.

## 3.2 «DolWin2» cluster wake on 11 October 2018

The «DolWin2» cluster is approximately 55 km upstream of GT I in southeasterly direction. We regularly have indications in our measurements for wakes from the cluster approaching GT I in stably stratified situations. Here we present a situation in autumn 2018 with a change of stability over the course of the day. We present a single lidar scan and an averaged lidar

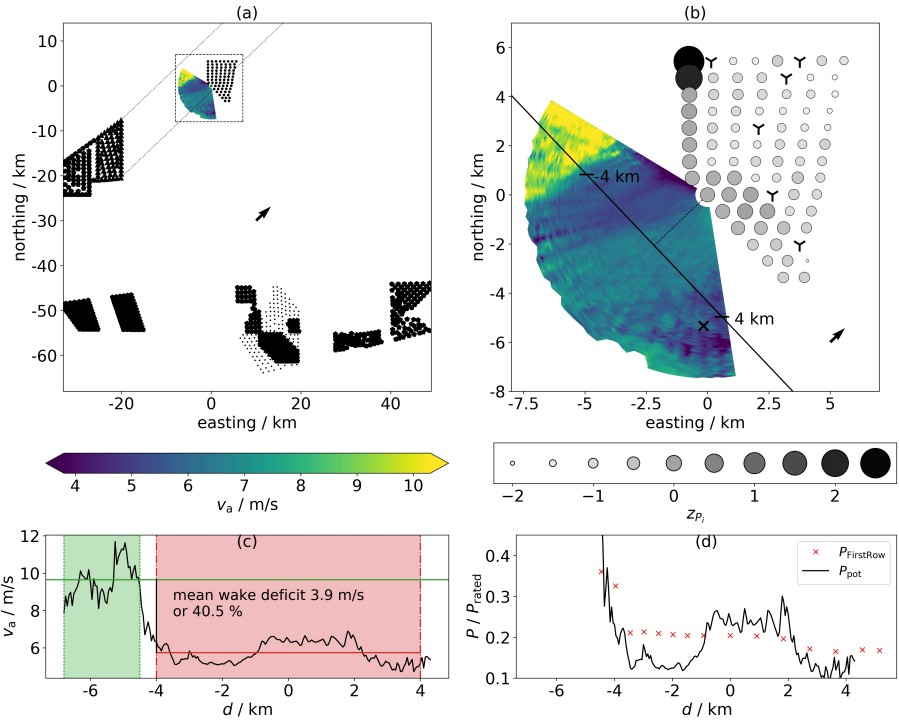

**Figure 6.** Lidar scan (scenario B from Table 2) on 6 February 2019 16:58 - 17:01: (a) Overview of the situation in the German Bight with lines parallel to the wind direction retrieved from the lidar scan from the corners of the upstream wind farm cluster «BorWin». Lidar wind speed is colour coded (left colour bar). (b) Close view of the lidar wind field and the wind farm GT I. The z-score of the not curtailed wind turbines's power in the current 10 minute interval (16:50 - 17:00) is grey coded (right colour bar), curtailed or not operating turbines are marked (Y-shaped marker). Markers scale with $z - z_{min}$. Turbine numbers to calculate the z-score as in Figure 5. The substation «Hohe See» (×) is marked. The solid line marks a virtual wake cut 3000 m upstream of turbine GT58 on which the wind field is evaluated and drawn in (c). Areas of wake and free stream are shaded manually, the resulting wake deficit is stated. (d) Available power on hub height along the wake cut from (b) together with the power achieved from the upstream turbines in GT I with their positions projected to the wake cut.

wind field from a period with low variation in the wind direction in stable stratification. A complementary SAR scan from the morning of the day during weakly unstable stratification is available as well and analysed here.

### 3.2.1 Meteorological conditions

In Figure 8 we plot the measured meteorological quantities on 11 October 2018. Since the lidar for measurements of wind
5  speed and direction and the data of air temperature, pressure and humidity at TP height were not available during the whole day we added the mesoscale data from the New European Wind Atlas (NEWA) and measurements from the nacelle of turbine GT58 to the plots. A cyclone southwest of Iceland and a strong high pressure area over Russia dominated the weather during the day. The North Sea was positioned in the warm sector of the cyclone between the cold front over the UK and the warm

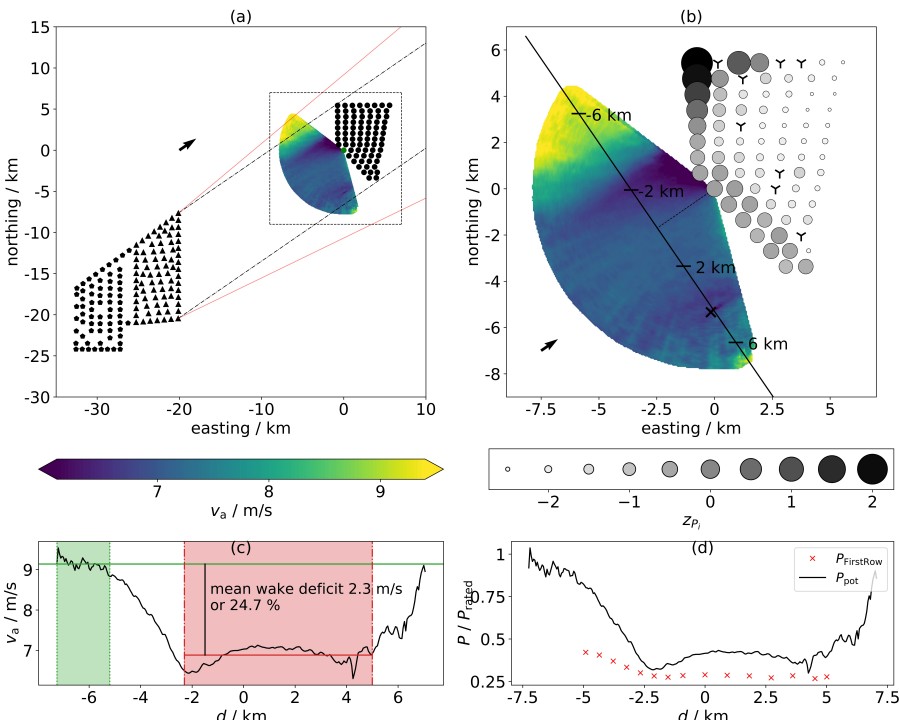

**Figure 7.** as in Figure 6 but averaged over 60 consecutive lidar scans (scenario B) corresponding to a period of 157 minute (13:35 - 16:12), power data averaged over 170 minutes (13:30 - 16:20). Red lines in a) indicate minimal and maximal wind directions within the averaging interval.

front spanning from Iceland to Norway. Southeasterly winds prevailed in the southern North Sea raising the air temperature in GT I between 12:00 and 14:00 above the temperature of the still quite warm North Sea (approximately $16\,°C$) stabilizing the boundary layer. In the morning a shallow (weakly) unstable boundary layer of some hundred metres height occurred because the surface layer over land cooled down during the night to temperatures below sea surface temperature and moved with
5  the prevailing flow over the sea. Aside the stability obtained from NEWA (weakly unstable) and the nacelle measurements (unstable) this finding is further supported by temperature profiles sounded with radiosondes at the stations in Bergen (nr. 10238) and Ekofisk (nr. 1400) the same day. A weak inversion with temperatures of approximately $13.5\,°C$ up to $300\,m$ height appears in the profile at Bergen, 04:00 UTC, with a stronger temperature inversion above. At the Ekofisk site the temperature profile at 11:00 UTC shows a similar behaviour with the upper inversion being less pronounced and sunken to approximately
10  $230\,m$ height. This allows for dry adiabatic convection up to heights between $200\,m$ and $300\,m$ for the prevailing sea surface temperature.

  We found a good general agreement between the NEWA data and the values measured in the wind farm. Especially, the derived stability parameter $\zeta$ agrees well. For the differences in the other quantities the different reference heights have to

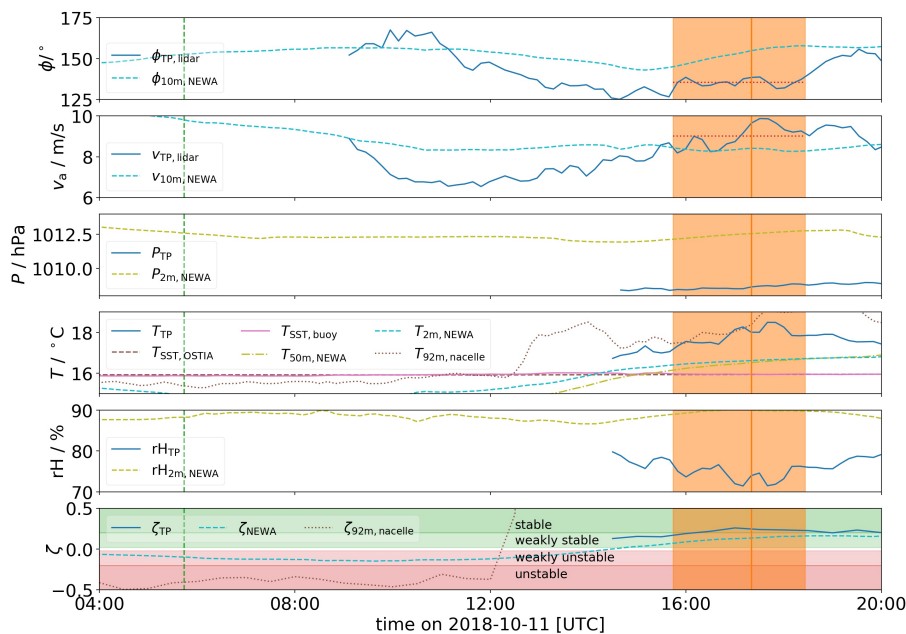

**Figure 8.** Meteorological data at the lidar location (turbine GT58) on 11 October 2018. Top to bottom: wind direction $\phi_{\text{TP,lidar}}$, wind speed $v_{\text{TP,lidar}}$, air pressure $P_{\text{TP}}$, air temperature $T_{\text{TP}}$, sea surface temperatures $T_{\text{SST,OSTIA}}$ and $T_{\text{SST,buoy}}$, relative humidity $\text{rH}_{\text{TP}}$ and the dimensionless stability parameter $\zeta_{\text{TP}}$ on the height of the TP of GT58 (24.6 m MSL). Since the measurements are not available during the whole day we added the 10 m wind speed $v_{\text{10m,NEWA}}$ and direction $\phi_{\text{10m,NEWA}}$, 2 m and 50 m temperature $T_{\text{2m,NEWA}}$ and $T_{\text{50m,NEWA}}$ and the stability parameter $\zeta_{\text{NEWA}}$ from the NEWA data set (c.f. Witha et al., 2019) as well as the temperature $T_{\text{92m,nacelle}}$ and the derived stability parameter $\zeta_{\text{92m,nacelle}}$ on hub height of turbine GT58. Measurement times are marked as follows: vertical dashed line: SAR image (Figure 9), vertical solid line: single lidar scan (Figure 10), shaded interval: averaged lidar wind field (Figure 11). Mean wind speed and direction in the averaged lidar interval are marked by red horizontal dotted lines.

be considered. Half-hourly values of wind speed and direction from the NEWA data are not expected to cover small scale fluctuations and to perfectly match a local measurement.

### 3.2.2 SAR wind data

Figure 9a draws the wind field from the Copernicus satellite Sentinel-1A, which passed the German Bight in the morning of 11 October 2018 as an overview of the wind field in the region between GT I and the «DolWin2» cluster. The stratification during the SAR snap shot was weakly unstable. Wakes of the «Gemini», «DolWin1» and «DolWin2» clusters with lengths of at least 20 km, 40 km and 55 km are clearly visible. The wake originating in the «DolWin2» cluster splits into two parts generated by «Gode Wind 1+2» (GW) and «Nordsee One» (N1), c.f. Figure 1. The GW wake extends far downstream until it hits the wind farm GT I after approximately 55 km. Further downstream a merged wake of the «DolWin2» cluster and GT I can be observed extending out of the visible range after approximately 30 km. All wakes have the approximately same width as the generating

cluster and become narrower downstream.

The virtual wake cut 9000 m upstream of GT58 reveals regions of different influence (Figure 9c). On the southwest side of the cut we see a region of undisturbed flow ($d \approx -15$ km, $d$ is the distance on the wake cut from Figure 9c) with wind speeds decreasing towards northeast. The deficit between $-5$ km $< d < 0$ km originates in the wake of the wind farm N1 followed by the stronger deficit at $0$ km $< d < 10$ km of the GW wind farm. This wake deficit centrally hits GT I and affects its power production. Further east the wind speed remains approximately constant until it rises from $d > 20$ km due to regional differences in the wind field. Regarding the marked wake and free flow regions in Figure 9c we observe a wake deficit of $0.6$ m s$^{-1}$ or 7.2 % in the SAR wind speed for the «DolWin2» wake in 10 m height.

Differently from the wake situation of the «BorWin» cluster (Section 3.1) the wind farm GT I is affected by the DolWin wake centrally, therefore we do not observe separated regions of power production within the farm. Nevertheless, the outer turbines on the western and northeastern corner of the wind farm produce significantly more power (2.6 and 1.7 $\sigma_P$ above average) than the average of the upstream row (Figure 9b). Looking at the potential power on hub height calculated from the virtual wake cut (Figure 9d) we find the increased power to result from the higher wind speeds at the sides of the «DolWin2» wake deficit. This highlights the effect of the wake on the power production even in weakly unstable conditions.

### 3.2.3 Lidar wind fields

In Figure 10 we show a single lidar scan of the flow southwest of GT I. The stratification during the scan was stable (Figure 8). We do not observe a sharp transition from wake to free flow regions like for the «BorWin» wake (Figure 6) but a steady decrease in wind speeds southwest to northeast similar to the «DolWin2» wake situation we found in the SAR data from the same morning in weakly unstable stratification (Figure 9). Three more wakes appear in the wind field, one originating from a ship close to GT I, another one from the OSS «Hohe See» ($\times$) and the third from the platform «BorWin gamma» ($+$). The latter wake extends at least 9 km downstream.

The virtual wake cut (Figure 10c) highlights the different flow regions with lower wind speeds near GT I. The «Hohe See» OSS wake is located at $d \approx -4$ km and the «BorWin gamma» wake between $-6$ km $< d < -5.5$ km. The wake deficit of the «DolWin2» cluster amounts to 3.3 m s$^{-1}$ or 26.4 %. Comparing the potential power in the wind field with the power produced by the turbines of the upstream row we find most turbines producing approximately rated power (Figure 10d). The potential power in the west of the wind farm is slightly lower than the power of the upstream turbines. Even though during this lidar scan with high wind speeds the wind farms power is not influenced by the «DolWin2» wake due to the turbines curtailing power production above rated speed, we find clear indications for wake effects with reduced wind speeds at the position of GT I 55 km downstream the DolWin 2 cluster.

Figure 11 highlights the steadiness of the «DolWin2» wake situation on 11 October 2018. We averaged 16 consecutive lidar scans in a period of approximately 162 minutes (15:44 to 18:26, cf. shaded interval in Figure 8) with a relatively constant wind direction. As for the single lidar scan we observe the same behaviour in the wind field with a wind speed decreasing along the virtual wake cut from southwest to northeast. The wake deficits of the «Hohe See» OSS and «BorWin gamma» are clearly visible in the averaged wind field (Figure 11c). The relative wake deficit of the «DolWin2» cluster is similar for the single and

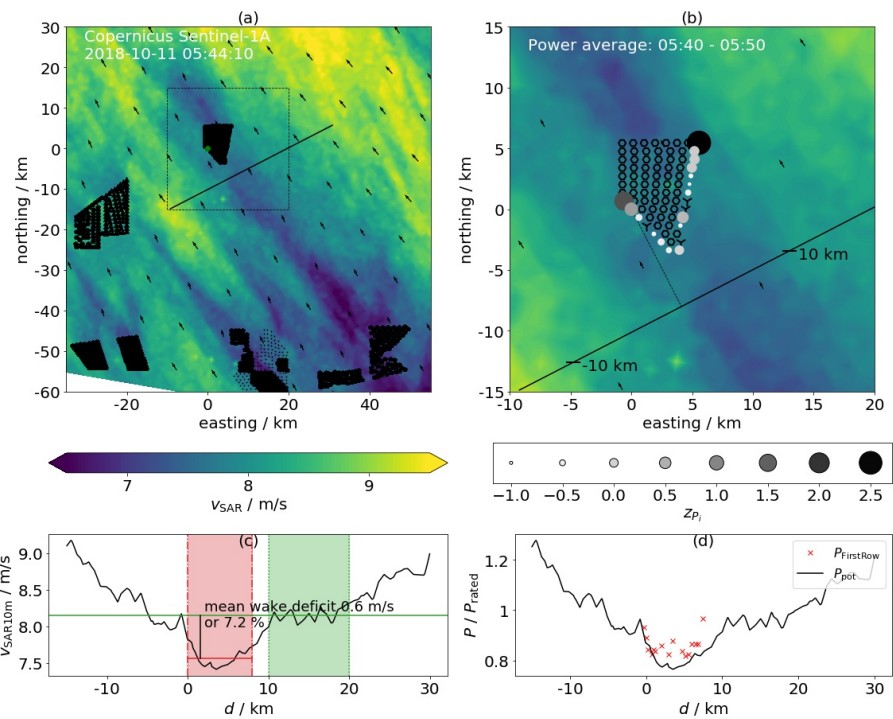

**Figure 9.** Sentinel-1A Ocean Wind Field (Copernicus Sentinel data [2018]), measurement taken 11 October 2018 05:44:10 UTC. We show power data of the upstream turbines in the interval 05:40 - 05:50, as in Figure 5, positions of downstream turbines are marked (hexagon). In d) we added an offset of $2.0\,\mathrm{m\,s^{-1}}$ to the SAR wind speeds on the virtual wake cut 9000 m upstream GT58 before we transferred them to hub height and calculated the potential power. Numbers of considered upstream turbines to calculate the z-score are 8, 15, 22, 29, 36, 43, 50, 68, 72, 80, 79, 76, 73, 64, 58, 51.

the averaged lidar scans (Figure 10). Since the average wind speed within the averaging period is smaller than that at the time of the single scan (Figure 8) the absolute deficit is smaller, too. The course of the potential power in the wind field (Figure 11d) is continued by the power of the upstream rows turbines. The wake effect of the «DolWin2» cluster on the power of GT I is evident. The potential power in the wind about 4 km southwest of the wind farm reaches rated wind speed.

## 4   Discussion

We found evidence of cluster wakes in form of wind speed deficits with clear transition regions between slower wake flow and faster undisturbed flow in many lidar scans upstream GT I for all neighbouring wind farm clusters in southeasterly to westerly wind directions, namely the «DolWin2» (approximately 55 km), «DolWin1» (approximately 42 km), «Gemini» (ap-

10   proximately 54 km) and «BorWin» (approximately 24 km) clusters. In some of the cases with available large-area SAR wind

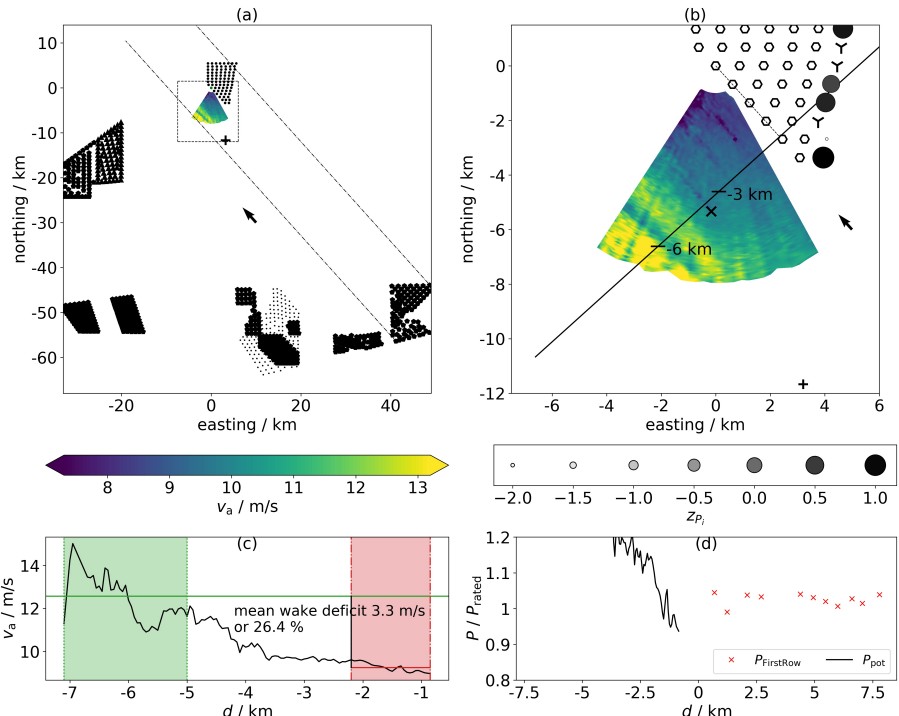

**Figure 10.** Lidar measurement (scenario A) of the wake of the «DolWin2» cluster on 11 October 2018 17:16 - 17:20, power data of upstream turbines 17:10 - 17:20, as in Figure 6. Downstream turbines positions marked (hexagon). Turbine numbers to calculate the z-score are 8, 15, 22, 29, 36, 43, 50, 68, 72, 78, 80. Additionally we marked the converter platform «BorWin gamma» (+).

data these alternative measurements supported the lidar cluster wake measurements. Power deficits in the wind farm agree with the wake regions found in lidar and SAR data. In this paper we presented two exemplary wake cases, one for the «BorWin» cluster 24 km upstream and one for the «DolWin2» cluster 55 km upstream, both wake effects occurred steadily over more than 2.5 hours and influenced the power production of GT I. We found cluster wakes mainly for positive values of the stability parameter $\zeta$ (stable stratification) but as well as for $\zeta$ slightly below zero (weakly unstable stratification, shallow boundary layer).

## 4.1 Influence of cluster wakes on power production of far downstream wind farms

The effect of cluster wakes on the operation of far downstream wind farms has not been investigated before. Nygaard and Hansen (2016) report about short distance effects in the power production of wind farms in direct vicinity (3.3 km gap) based on SCADA analysis. Nygaard and Newcombe (2018) analyse a cluster wake at hub height up to 17 km downstream a wind farm with dual Doppler radar from the coast. Platis et al. (2018) find long reaching wake effects (wind speed difference of more than $0.1\ \mathrm{m\,s^{-1}}$ considered as wakes) up to 55 km downstream in flight measurements but could not analyse their impact

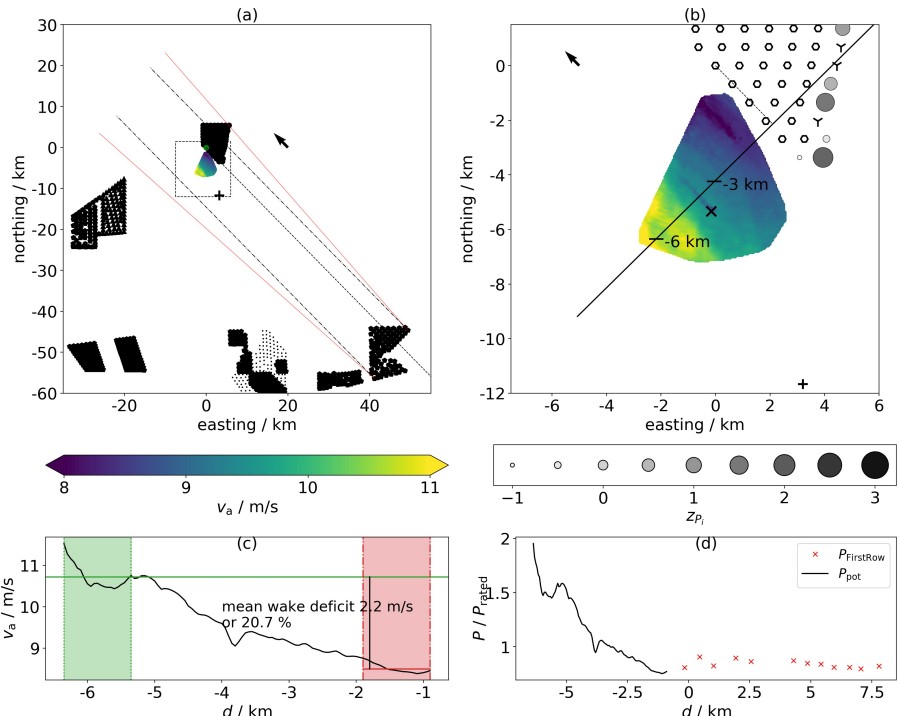

**Figure 11.** Wake of the «DolWin2» cluster on 11 October 2018 as in Figure 6 but averaged over 16 consecutive lidar scans (scan scenario A) in a period of 162 minutes (15:44 - 18:26), power data of upstream turbines averaged over 170 minutes (15:40 - 18:30), downstream turbines marked (hexagon). Turbine numbers to calculate the z-score are 8, 15, 22, 29, 36, 43, 50, 68, 72, 78, 80, 79.

on distant wind farms. Here, our findings from combined satellite SAR and lidar measurements of cluster wakes existing over distances of up to 55 km downstream agree with the observation of Platis et al.. Additionally, we confirm the assumption of negative effects of cluster wakes on the power production of a far downstream wind farm.

The evidence of the wake influence on wind farm power is obvious for the «BorWin» case where we find a clear distinction
5    of wake and free stream in the lidar and SAR wind measurements agreeing with the findings of Platis et al. who present a wake situation with a high wind speed gradient at one side of the cluster wake. In the «BorWin» case this edge of the wake continues in a separation of the wind farm turbines power production (Figures 5, 6, 7). In the «DolWin2» case we could argue whether the higher power of the outer turbines (Figure 9b) results from flow effects at the farm corners leading to higher turbine efficiencies as found by Barthelmie and Jensen (2010) but the comparison of the potential power in the inflow with the turbine
10   power (Figure 9d) reveals a good agreement suggesting that at least most of the effect originates in the wake affected inflow conditions with the highest deficit reducing the power of the central turbines while the outer turbines profit from higher wind speeds at the sides of the wake.

Wakes are expected to exist far downstream in stable stratifications but to recover much earlier in the unstable case. Platis et al. (2018) report about 41 measurement flights (24 × stable, 12 × unstable, 5 × neutral stratification) and find evidence for cluster

wakes in stable boundary layers 55 km downstream while the furthest evidence in an unstable case is found 10 km downstream. In our lidar measurements we find the most pronounced cluster wakes in stable situations supporting these findings. But we have evidence for far reaching wakes in neutral and weakly unstable conditions, too. All lidar measurements we present in this work were measured in stable situations but the SAR image of the «DolWin2» case (Figure 9) was taken earlier the

same day in a shallow weakly unstable boundary layer with cluster wakes appearing downstream of many clusters. Vertical momentum transport was possible in lower heights but was hindered by an inversion appearing at approximately 200 m to 300 m. The rotor area of the GT I turbines extends up to 150 m height. The «DolWin2» wake reaches 55 km downstream until it hits the wind farm GT I where the power production of the upstream row turbines follows the potential power calculated from the inflow SAR wind. This finding proves the existence of long reaching cluster wakes and their influence on power

production of far downstream wind farms even in cases with weakly unstable stratification. In future work we plan to publish an analysis of the whole data set of the, at the time of writing, still ongoing lidar measurement campaign focusing on wakes in unstable conditions. Nevertheless, the «DolWin2» case highlights the necessity to carefully characterize the boundary layer for stability analysis, since the unstable stratified layer in the boundary layer could be thin and limited by an inversion just above temperature measurement height and still within the rotor area.

In addition to the influence of a cluster wake on the wind farm GT I we still observe inner farm wake effects (Figures 5, 6) with decreasing power production downstream. Cluster wake and wind turbine wakes in the farm overlap. This supports the assumption of the cluster wake being a region of reduced wind speeds with no special characteristics of the original single turbine wakes remaining. We do not perform turbulence analysis comparing cluster wake turbulence to free flow turbulence in this study. Platis et al. (2018) report a slender wake of increased turbulent kinetic energy (TKE) originating in one corner of

the cluster. It was aligned with a stronger horizontal wind speed gradient at the border of the wake. The TKE was reduced in the wake deficit due to the lower wind speeds.

The influence of cluster wakes on the current power production of downstream wind farms could not easily be related to their influence on the annual energy production (AEP). To achieve this, a detailed assessment of the total influence during at least one year has to be conducted using e.g. validated wind farm parametrizations in mesoscale models. The local distribution of

wind speed, direction and atmospheric stability has to be considered as well as farm and cluster geometries.

In many wake cases the wind speed in the wake deficit still exceeds rated wind speed of the downstream turbines without an effect on their power production. If the upstream cluster's turbines operate in wind speeds above rated speed their thrust coefficient $c_T$ decreases additionally resulting in reduced wake deficits. We expect the total influence of cluster wakes on AEP to be smaller than wake effects from neighbouring wind farms (c.f. Nygaard and Hansen, 2016) due to cluster wake

recovery and a smaller wake influenced wind direction sector. Our findings do not question wind energy utilisation in any kind. Nevertheless, a detailed assessment of the influence of cluster wakes on AEP of downstream wind farms during their whole operational life time considering all planned wind energy activities in the region should be conducted in the future. This can improve power production, offshore resource assessment and consequently reduce the uncertainties in financing large offshore wind projects especially in regions with a high level of (planned) wind energy utilisation. Therefore, further research is

necessary to validate wind farm parametrizations in mesoscale weather models with appropriate wake, power and atmospheric

measurements. Especially the influence of atmospheric stability on cluster wake recovery has to be investigated.

Aside from influence on power the effect on additional wind turbine loads can be relevant. We did not perform analysis of the turbulence in the wake in this study or load simulations on wind turbines affected by far cluster wakes. Since we find sharp edges between wake flow and free stream continuing in the wind farms power production (Figure 6) future research should analyse turbine loads dependent on the cluster wake dynamics e.g. when a turbine on the wake border has to speed up and down fast caused by cluster wake dynamics.

## 4.2   Cluster wake characteristics

Wind turbines are sensitive to the wind conditions over a wide range of heights defined by the swept rotor area. Therefore, the investigation of cluster wakes should cover the whole vertical wind profile at least from lower to upper tip height. Satellite SAR measurements at the sea surface are typically transferred to 10 m height. Platis et al. (2018) investigates cluster wakes at hub height with a research aircraft in stable stratification while Siedersleben et al. (2018b) additionally presents measurements in five different height levels (60 m, 90 m, 120 m, 150 m, 220 m) from the same flight revealing wake deficits in all regarded levels. This highlights a vertical expansion of the wake far above the rotor area (upper tip height: 150 m). We find evidence for cluster wake effects in SAR images (roughness measurement on the sea surface, interpolation to 10 m above sea level), lidar measurements ($\approx$24.6 m above MSL, 67.0 m below hub height and 9.0 m below lower blade tip height) and from the turbines power production (rotor swept area spans from 33.6 m to 149.6 m above MSL). A quantitative comparison of the measured wake strengths is not possible with our data due to the very different type of the measurements. Nevertheless we obtain evidence for wake effects in the boundary layer from the sea surface to the upper tip height 24 km and 55 km downstream agreeing with the observed vertical wake extension closer to the generating cluster presented by Siedersleben et al. (2018b). For a future campaign we suggest the assessment of the development of the atmospheric boundary layer from the inflow through a cluster and in the cluster wake by means of e.g. lidar profilers, lidar range height indicator scans (RHI) or flight measurements for a better understanding of cluster wake development and recovery.

All previous investigations of cluster wakes with satellite SAR suffer from the fact, that just one snap shot of the wake is available for a given situation and no wake dynamics or their steadiness could be analysed. Nygaard and Newcombe (2018) investigate a cluster wake at hub height up to 17 km downstream a wind farm with dual Doppler radar from the coast and present a one hour average wake field. The aircraft measurements performed by Platis et al. (2018) cover the whole area of the wake along the flight path taking several hours indicating a constant behaviour of the wake. We find steady wake conditions in both presented examples for more then 2.5 hours in the lidar data supported by the corresponding power data. This proves the existence of steady wake effects with a steady influence on the downstream wind farm for constant wind directions. Wake cases with changing wind directions are much harder to analyse since the wake just shortly influences the farm and will probably not even be detectable in wind measurements.

We did not find any evidence for single wind turbine wakes in the lidar inflow measurements of GT I. This is supported by the results by Nygaard and Newcombe (2018) who present dual-Doppler radar cross stream flow cuts through a cluster wake at different downstream distances with disappearing signatures of the single turbines from 6 km downstream (unknown stability).

The shapes of the wakes we find could give further hints on the wake recovery process. While shorter wakes (here i.e. from the «BorWin» cluster, Figure 5) are as wide as the generating cluster wakes originating further away like from the «Gemini» cluster often appear narrower in the lidar measurements as if they already recovered from the sides or if the whole wake has widened with a resulting decrease in maximum wake deficit. This is supported by the shapes of the wakes seen in the SAR wind data in Figure 9b where the highest wake deficits are narrower further downstream. A detailed analysis of this effect is difficult due to changes in the mesoscale wind field and wakes of neighbouring clusters overlapping with the cluster wake.

The width of the transition region between free flow and wake seems to (at least partly) depend on the downstream position of the wake. In the «BorWin» wake we sometimes find high wind speed gradients at the wake's border about 20 km downstream (Figure 6) while in the «DolWin» wake 50 km downstream the transition region was several kilometres wide (Figure 10).

The longevity of wakes in stable conditions is further supported by the investigation of two different converter platform wakes in our lidar measurements ranging at least 9 km downstream in one case (Figure 10). Platform wakes have been observed before, e.g. Chunchuzov et al. (2000) reported a more than 60 km long wake of a 164 m tall offshore platform in very stable atmospheric conditions analysed with satellite SAR measurements. We did not investigate the effect of the wakes of wind farm converter platforms on the power of neighbouring or distant wind turbines but expect it to be fairly small compared to a wind turbine wake due to the lower heights and smaller cross sections of the platforms.

## 4.3   Cluster wake monitoring

Due to the large areas the cluster wakes take up their investigation was mainly based on long ranging remote sensing techniques. Satellite SAR covers large areas and has been widely used to analyse cluster wakes (Hasager et al., 2015). Our analysis adds the potential power as a computed local quantity to the SAR analysis (Figure 5d) confirming the wake shape acquired by SAR with turbine power data. This is another hint for the ability of satellite SAR to resolve flow structures agreeing with the findings of Schneemann et al. (2015) who compared structures in concurrent SAR and lidar measurements indicating the general ability of SAR to resolve flow structures with the size of a few hundred metres.

Cluster wakes have not been measured with long range lidar. With an achievable maximum range of 10 kilometres with compact devices lidar seemed not to be appropriate to measure far cluster wakes behind a wind farm. We used lidar to measure incoming far cluster wakes. As opposed to SAR lidar allows for continuous measurements with scan repetition times in the order of a few minutes (2.5 min and 10 min here). In some cases the lidar results are clear (e.g. Figure 6) but in other cases it is difficult to interpret whether the wind field is influenced by a wake or not. Here, satellite SAR, when available, proves very useful to interpret wind monitoring by lidar offering the possibility to regard the lidar wind field in a wider context (e.g. the «DolWin2» case, section 3.2). Nevertheless, absolute wind speed measurements by satellite SAR are comparably imprecise. For the comparison of the shapes of the potential power in the inflow with the turbines power we had to correct individual offsets in the SAR wind speeds within the given measurement accuracy. Schneemann et al. (2015) had to correct for an offset in SAR winds, comparing it with lidar, as well. This inaccuracy could be possibly reduced by a SAR analysis tuned to the special case. We did not perform SAR wind calculations ourselves but used already processed wind data.

The analysis of SCADA data on power losses due to cluster wakes without additional flow information from e.g. remote sensing

is difficult since obvious gradients in wind farm power (Figure 6) due to cluster wakes are rare and not exactly stationary (e.g. washed out transition region in averaged lidar wind field, Figure 7b). In the «DolWin2» case (Figure 9) it is hardly possible to judge on the contributions of wake effects and effect of higher turbine efficiency at the farm's corners (Barthelmie and Jensen, 2010) on the higher power of the turbines at the eastern and western corner of the farm.

For future research on cluster wakes and their influence on power generation we propose a combination of different measurement techniques complementing with their advantages, namely satellite SAR, long range lidar and flight measurements (aircrafts and drones). Doppler radar and non-compact lidar systems offering ranges larger than 15 km are available, but have not been deployed in offshore wind farms so far due to high costs and technical hurdles in the deployment, orientation and operation of the container-size systems on offshore structures.

Another important aspect of measurements from offshore platforms like transition pieces of offshore wind turbines to be considered is platform movement and the resulting errors in measurement locations. We found platform tilts of up to $0.1°$ due to turbine thrust depending on wind speed and direction using the method of sea surface levelling (Rott et al., 2017). This value might be even higher for turbines on a today commonly used monopile foundation compared to the tripod foundation used in GT I. With increasing measurement ranges the location error in the measurements further grows.

## 5  Conclusions

This paper investigates the question, whether offshore cluster wakes have an influence on power generation of far downstream wind farms considering atmospheric stability. Therefore we analysed two different cases of 24 km and 55 km long cluster wakes approaching the 400 MW offshore wind farm «Global Tech I» (GT I) by means of satellite SAR measurements, lidar wind monitoring as well as analysis of atmospheric stability and GT I power production.

Long range Doppler lidar supported by satellite SAR proves as a good combination for cluster wake measurements with the lidar providing accurate wind speed monitoring over long periods and SAR contributing with large-area wind fields for the overall picture.

We find that long distance wake effects of a wind farm cluster exist at least 55 km downstream in stable and weakly unstable stratification. They persist for more than 2.5 hours. During this measurement period the average wake deficits are $2.3 \, \mathrm{m\,s^{-1}}$ or

25 % approximately 24 km downstream and $2.2 \, \mathrm{m\,s^{-1}}$ or 21 % approximately 55 km downstream. Single lidar scans (2.5 min duration) reveal stronger wake deficits of up to $3.9 \, \mathrm{m\,s^{-1}}$ or 41 % approximately 24 km downstream.

Clear transition regions like edges in the wind separate wake and free flow 24 km downstream and continue in the affected wind farm splitting it in regions of higher power in undisturbed flow and reduced power in the wake deficit. Free flow turbines produce more then two standard deviations $\sigma_P$ more then the average of the upstream turbines.

This contribution proves the existence of steady power reductions in a far downstream wind farm caused by cluster wakes. We encourage further investigations on far reaching wake shadowing effects for optimized areal planning at sea and reduced uncertainties in offshore wind power resource assessment.

## Appendix A:  Calculation of virtual potential temperatures

We derived the virtual potential temperature used in section 2.4 from the available measurements on the TP. We adapted the following methodology mainly from Etling (2008). We need

- $R_d = 287 \; \frac{\text{J}}{\text{K} \cdot \text{kg}}$ (specific gas constant of dry air)

- $R_v = 461 \; \frac{\text{J}}{\text{K} \cdot \text{kg}}$ (specific gas constant of water vapour)

- $\epsilon = \frac{R_d}{R_v} = 0.622$ (ratio between the specific gas constants for dry air $R_d$ and water vapour $R_v$)

- $\kappa_P = 0.286$ (Poisson constant in dry air).

The saturation vapour pressure dependent on the temperature follows from the Magnus equation

$$e_s(T)[\text{Pa}] = 100.0 \cdot 6.1 \cdot 10^{\left( \frac{7.45 \cdot (T[\text{K}] - 273.15)}{T[\text{K}] - 38.15} \right)}. \tag{A1}$$

The partial pressure of water vapour in the air dependant on the relative humidity rH reads

$$e = \text{rH} \cdot e_s / 100.0 \tag{A2}$$

while the mixing ratio is

$$r_v = \epsilon \cdot \left( \frac{e}{p - e} \right). \tag{A3}$$

With the specific humidity

$$q = \frac{r_v}{1 + r_v} \tag{A4}$$

and the potential Temperature

$$\Theta = T \left( \frac{100,000 \; \text{Pa}}{p} \right)^{\kappa_P} \tag{A5}$$

we approximate the virtual potential temperature

$$\Theta_{\text{v}} = \Theta \cdot (1.0 + 0.61 \cdot q). \tag{A6}$$

While the virtual potential temperature at the TP $\Theta_{\text{v,TP}}$ could be derived directly from the available measurements we assume the relative humidity and the air temperature directly above the sea to be $\text{rH}_0 = 100 \, \%$ and $T_0 = T_{\text{SST}}$ respectively to derive the virtual potential temperature at sea level $\Theta_{\text{v,SST}}$. Furthermore we calculate the air pressure at sea level

$$p_0 = p_{\text{TP}} \cdot \left( \frac{T_{\text{SST}} - \gamma \cdot z_{\text{TP}}}{T_{\text{SST}}} \right)^{\frac{-g}{\gamma R_d}} \tag{A7}$$

assuming a polytrop atmosphere and using the air temperature gradient

$$\gamma = \frac{T_{\text{SST}} - T_{\text{TP}}}{z_{\text{TP}}}. \tag{A8}$$

*Data availability.* Lidar data is not published and could be made available on request. GT I SCADA data is confidential and therefore not available to the public. SAR wind data is available from https://scihub.copernicus.eu/. Hourly power data for several wind farms is available from https://www.energy-charts.de/. The «New European Wind Atlas» is published at https://map.neweuropeanwindatlas.eu/. The OSTIA data set could be obtained from http://marine.copernicus.eu/ and radiosonde soundings are available at www.meteociel.fr or http://weather.uwyo.edu.

*Author contributions.* Jörge Schneemann conducted and supervised the measurement campaign, designed the research, performed the data analysis, made the figures and planned and wrote the paper. Andreas Rott and Martin Kühn contributed to the research with intensive discussions and added to the paper with conceptual discussions and internal review. Martin Dörenkämper advised on the meteorological parts, participated in the conception of the paper and did an internal review. Gerald Steinfeld performed parts of the stability analysis and intensively reviewed the manuscript.

*Competing interests.* The authors declare no conflict of interest.

*Acknowledgements.* We performed the lidar measurements and parts of the work in the framework of the research project "OWP Control" (FKZ 0324131A) funded by the German Federal Ministry for Economic Affairs and Energy on the basis of a decision by the German Bundestag. We acknowledge the wind farm operator Global Tech I Offshore Wind GmbH for providing SCADA data and their support of the work. Furthermore, we thank the European Space Agency (ESA) for making the Sentinel-1 data of the Copernicus program available. Thanks to Met Office for making the OSTIA data set available. We acknowledge the NEWA consortium for providing access to the «New European Wind Atlas». Special thanks to Stephan Voß for his work on the measurement campaign and the picture from Figure 3.

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
