# Peer review of "Cluster wakes impact on a far distant offshore wind farm's power"

_Wind Energy Science, 2019_

## Referee Comment (RC1) · Anonymous Referee #1 · 27 Aug 2019

Reviewers Comments

Title: Cluster wakes impact on a far distant offshore wind farm's power Authors: Jörge Schneemann, Andreas Rott, Martin Dörenkämper, Gerald Steindfeld, Martin Kühn DOI: 10.5194/wes-2019-39

General comments

The authors did an excellent job in combining lidar data with SAR data. This approach is novel and the observations resulting from this approach make an important contribution to the ongoing research in the field of mesoscale wakes of offshore wind farm clusters and their impact on the power production. Additionally, the manuscript is clearly written and structured. However, I have one minor comment (see below) that can be

considered as minor, but at the same time this comment is substantial in nature and must be done.

The authors present two case studies whereby one is characterized by stable conditions and the second one by weakly unstable and stable conditions. However, the stability criterion they applied to define the stratification of the atmosphere can be misleading as they only consider the atmosphere below 24.6 m MSL to obtain the stability of the atmosphere. In contrast, the aircraft observations of Platis et al. (2018) and analytical models (e.g. Emeis (2009)) reveal that the atmosphere at rotor height and above are of major importance when defining the stability of the atmosphere when considering wakes of wind farms. The state of the atmosphere above the rotor drives the vertical turbulent momentum flux, that in turn drives the recovery of the wind deficit. Consequently, using a bulk Richardson number using measurements at sea surface and at 24.6 m are independent of the atmosphere above 24.6 m. This point is totally missing in the discussion, especially in the paragraph P20L8-19. The authors claim that they observed a wake with a length of 55 km in weakly unstable conditions, but they don't mention the possibility that an inversion above 24.6 m could have hindered vertical momentum transport from above. For example, in the case study presented in Siedersleben et al. (2018), an inversion was present above hub height but the atmosphere was weakly unstable stratified below hub height in the morning hours. Therefore, the authors should at least mention the possibility of stable conditions above hub height, otherwise the results presented in this study are misleading. Depending on the motivation of the authors they could also check nearby soundings taken at the shore upwind to get an idea of the atmosphere above rotor height.

Other minor comments

P4L16: Why z-score, what advantage does this method have. Please comment on that!

P5L12: Figures 2 and 1 should be Figures 1 and 2

P6L1: ... with different settings ... Why not are you not mentioning the names of the settings? -> ... with two different setting A and B as listed in Table 2.

P8L12: How did you derive the virtual temperature at the sea surface? Did you interpolate the pressure measurements take at 24.6 m to the sea surface and what humidity did you use?

P9L21-P10L5: Does the RMSE in wind speed and wind direction correspond to a quality flag smaller or equal than 2?

Figure Comments

Fig. 5: Letters indicating the orientation of the cross section would be very helpful.

Fig. 6, Fig. 7 and Fig. 9, Fig. 10: Again, letters indicating the orientation of the cross section would be helpful. Additionally, the cross section as indicated in Fig. 6b) seems to be longer than shown in Fig. 6d).

Fig. 5b) and Fig. 6b): What is the meaning of the rotor like looking icons?

Fig. 8): ... in the averaged lidar interval are marked by ?red? horizontal dotted lines?

Fig. 9): Why is the power output only shown for the front row turbines? Are the other turbines producing less than rated power and the z-score is, hence, omitted for these turbines?

Emeis, S.: A simple analytical wind park model considering atmospheric stability, Wind Energy, 13, 459–469, https://doi.org/10.1002/we.367, 2009.

Platis, A., Siedersleben, S. K., Bange, J., Lampert, A., Bärfuss, K., Hankers, R., Cañadillas, B., Foreman, R., Schulz-Stellenfleth, J., Djath, B., Neumann, T., and Emeis, S.: First in situ evidence of wakes in the far field behind offshore wind farms, Scientific Reports, 8, https://doi.org/10.1038/s41598-018-20389-y, 2018.

Siedersleben, S. K., Platis, A., Lundquist, J. K., Lampert, A., Bärfuss, K., Cañadillas, B., Djath, B., Schulz-Stellenfleth, J., Bange, J., Neumann, T., and Emeis, S.: Evaluation of a Wind Farm Parametrization for Mesoscale Atmospheric Flow Models with Aircraft Mea- surements, Meteorologische Zeitschrift, 27, 401–415, https://doi.org/10.1127/metz/2018/0900, 2018.

———————————————————

---

## Short Comment (SC1) · 2 Sep 2019

Dear authors, dear reviewer,

I found both the paper and the comment to the paper very interesting. I only wanted to mention that sounding data are available easily from https://www.meteociel.fr/observations-meteo/sondage.php?map=1, in particular Ekofisk (central North Sea). See examples for two timestamps relevant to the article in the folder https://www.dropbox.com/sh/ttm7w8ykk3i86m2/AADmzzAP2OqyBVs-tzEOpqn2a?dl=0.

This shows indeed that, in the second case, near the surface the water may be warmer than the air, while the temperature increases with height. This makes a singular situa-

tion, which I am guessing is not lasting very long, but which is worth exploring. See the file Overview_soundings_NorthSea_201810111200UTC.png in the folder mentioned above.

All the best Rémi Gandoin rga@c2wind.com

---

## Referee Comment (RC2) · Nicolai Gayle Nygaard (Referee) · 3 Sep 2019

The work presented in the manuscript combines SAR surface wind speed retrievals with scanning lidar measurements and SCADA data to characterise wakes from wind farm clusters and their impact on a wind farm in the German Bight. This is adding important new measurements and observations to a growing literature on cluster wakes.

The manuscript is well structured and the presentation of the results is clear and easy to follow. I definitely recommend that the paper is published. Only minor adjustments are necessary.

P. 2, line 9-10: this makes it sound like optimization of wind farm layouts to reduce wake effects is new. Of course, the industry has been doing this for many years. I suggest

slightly rephrasing to avoid this misinterpretation.

P.2, line 15: I suggest adding a reference to the work by Volker on the Explicit Wake Parametrisation (EWP), for example Geosci. Model Dev., 8, 3715–3731, 2015

P.3, line 13-14: maybe add that the wake and free flow regions are defined manually

P.4, line 16: clarify if the curtailment filter is based on a specific SCADA signal or derived from a combination of signals. This is of general interest to readers working with SCADA data. A reference to another paper describing similar filtering would also be sufficient

P.6, table 1: are the hub heights with reference to mean sea level?

P.6, line just above table 2: insert "the" between "in" and "beam"

P.7: how does the finite acquisition time combined with the scan rate "smooth" the lidar measurements? Is this similar to a spatial averaging? Please comment in the text

P.8, line 5: please add a description (and a reference if relevant) for the interpolation onto a regular grid. Also, it is not clear if the exclusion of grid points with less than 10 single scan contribution applies after interpolation or time averaging or both. Please clarify this

Eq. (3): in Stull (1988) the bulk Richardson number is defined in terms of the virtual potential temperature. The authors use the potential temperature. Please specify why this approximation is appropriate and does not introduce bias in the classification of stability. Furthermore, I have seen other papers where the temperature in the denominator of the bulk Richardson number is not the surface temperature, but the air temperature at the measurement height. This cannot change unstable to stable conditions, but it can shift the stability parameter closer or further from neutral conditions. Finally, I wonder if the lidar measurements used in equation 3 include measurements in the cluster wakes. If this is the case, the wind speed in the denominator of equation 3 will be too small, thus biasing the stability parameter away from neutral conditions

P. 9, line 13: which z is used to calculate the stability parameter. Is it z_TP? Please specify

P.12, figure 5: are the non-operating/curtailed turbines the one marked with the symbols that are not circles? This is not clear from the caption. The same symbol is used in figures 9-11, but is not referred to in the captions. Were most turbines standing still or curtailed in the second flow case?

p. 15, figure 8: I worry about the classification of weakly unstable conditions at the time of the 11 October 2018 SAR image. The stability is assessed entirely on a model. What is the uncertainty of this? Later in the same day, when meteorological measurements are available a large bias is seen between the mesoscale temperature and the measured temperature. This bias (if it also existed in the morning) could maybe change the stability classification from weakly stable to weakly unstable.

p. 16, line 30: can you comment on the expected AEP impact of the OSS platforms? I would expect it to be small, since the platforms are fairly low compared with the turbines and have a smaller cross-sectional area.

p. 17, line 3: at rated power OR above rated speed (as opposed to at rated speed)

p. 17, line 9: if the average wind speed is smaller the wind speed deficit should be larger not smaller due to the increase of the thrust coefficient at lower wind speeds. Or am I missing something?

p. 18, line 7: when saying that SAR mostly supports the lidar wake measurements, can you be more specific?

p. 21, line 14: I suggest this phrasing: Wind turbines are sensitive to the wind conditions over a wide range of heights defined by the swept rotor area.

p. 22, line 9: is the wind speed deficit region truly decreasing in width, or is it the region of a certain colour in the heat map that is shrinking? The increase of wake width is typically coupled with the decrease of the peak deficit.

p. 23, line 26: "then" should be "than"

---

## Author Comment (AC1) · 25 Sep 2019

Please find the answers to the reviewers comments in the supplement. A second pdf file highlights the changes in the manuscript.

Please also note the supplement to this comment: https://www.wind-energ-sci-discuss.net/wes-2019-39/wes-2019-39-AC1-supplement.zip

---

## Author Response (AR1)

**Cluster wakes impact on a far distant offshore wind farm's power**

Authors: JörgeSchneemann, Andreas Rott, Martin Dörenkämper, Gerald Steinfeld, Martin Kühn
DOI: 10.5194/wes-2019-39

**Author response to reviewer comments**

We would like to thank the two reviewers and Rémi Gandoin for their time and the constructive and helpful comments. Their input contributed to an improvement of the original manuscript. We addressed their feedback and reply to it point-by-point in the following. We highlight the changes in the text of the manuscript within a separate pdf file (LaTeX-Diff).

**Anonymous Reviewer #1**

**1. [Reviewer #1]** The authors present two case studies whereby one is characterized by stable conditions and the second one by weakly unstable and stable conditions. However, the stability criterion they applied to define the stratification of the atmosphere can be misleading as they only consider the atmosphere below 24.6 m MSL to obtain the stability of the atmosphere. In contrast, the aircraft observations of Platis et al. (2018) and analytical models (e.g. Emeis (2009)) reveal that the atmosphere at rotor height and above are of major importance when defining the stability of the atmosphere when considering wakes of wind farms. The state of the atmosphere above the rotor drives the vertical turbulent momentum flux, that in turn drives the recovery of the wind deficit. Consequently, using a bulk Richardson number using measurements at sea surface and at 24.6 m are independent of the atmosphere above 24.6 m. This point is totally missing in the discussion, especially in the paragraph P20L8-19. The authors claim that they observed a wake with a length of 55 km in weakly unstable conditions, but they don't mention the possibility that an inversion above 24.6 m could have hindered vertical momentum transport from above. For example, in the case study presented in Siedersleben et al. (2018), an inversion was present above hub height but the atmosphere was weakly unstable stratified below hub height in the morning hours. Therefore, the authors should at least mention the possibility of stable conditions above hub height, otherwise the results presented in this study are misleading. Depending on the motivation of the authors they could also check nearby soundings taken at the shore upwind to get an idea of the atmosphere above rotor height.

**[Authors]** This comment is similar to comment 12 of Reviewer #2. We refer also to the answer to this question.

In particular, we added temperature measurements from nacelles in GTI to characterize stability in the morning of 11.10.2018 and checked radiosonde soundings at Bergen and the Ekofisk site in the North Sea downstream of GTI (c.f. Figures 1 and 2 in the answer to question 12. of [Reviewer #2]). We found indeed an inversion, but it was situated in heights above hub height (nacelle measurements) and even above the rotor area (Bergen and Ekofisk soundings). Therefore, we stay with our statement of weakly unstable stratification in the very lowest layer below some 300 m where the turbines operate. In general, your comment to consider thermal stratification (or basically the height of the atmospheric boundary layer/internal boundary layer) additional to the use of stability coefficients is right and important. A higher boundary layer better supports the filling of the wake deficit with momentum from above than a low boundary layer. We added a statement on this fact in the discussion in the manuscript.

**2. [Reviewer #1]** P4L16: Why z-score, what advantage does this method have. Please comment on that!

**[Authors]** We choose the z-score since it better highlights differences in the power of the upstream turbines. The normalization with the std. deviation highlights the significance of the findings. To avoid distortion by inner farm wake effects we reference the z-score to the mean and std. dev. of the upstream turbines instead of using the whole farm.

**3. [Reviewer #1]** P5L12: Figures 2 and 1 should be Figures 1 and 2
**[Authors]** corrected

**4. [Reviewer #1]** P6L1: . . . with different settings . . . Why not are you not mentioning the names of the settings? -> ... with two different setting A and B as listed in Table 2.
**[Authors]** implemented

**5. [Reviewer #1]** P8L12: How did you derive the virtual temperature at the sea surface? Did you interpolate the pressure measurements take at 24.6 m to the sea surface and what humidity did you use?
**[Authors**] We specified and corrected the description of the derivation of the stability coefficient and added a short appendix to the manuscript describing the derivation of the virtual potential temperatures at the sea surface and on the height of the TP from the available data. We calculated the pressure at sea level using equation A7 added to the manuscript.

**6. [Reviewer #1]** P9L21-P10L5: Does the RMSE in wind speed and wind direction correspond to a quality flag smaller or equal than 2?
**[Authors]** We added the current Sentinel-1 Product Specification (c.f. reference Vincent et al., 2019) to the manuscript and specified the quality flag used as *owiWindQuality* and wrote down the definition (0: high quality, 1: medium quality, 2: low quality, 3: bad quality). The source for the RMSE for wind speed and direction does not link the error to the quality flag.

Figure Comments
**7. [Reviewer #1]** Fig. 5: Letters indicating the orientation of the cross section would be very helpful.
**[Authors]** We added labelled ticks on the cross sections in Figures 5, 6, 7, 9, 10 and 11 corresponding to the scale on the x-axis in subfigures c and d.

**8. [Reviewer #1]** Fig. 6, Fig. 7 and Fig. 9, Fig. 10: Again, letters indicating the orientation of the cross section would be helpful. Additionally, the cross section as indicated in Fig. 6b) seems to be longer than shown in Fig. 6d).
**[Authors]** We added labelled distance ticks, see answer above. Indeed, the cross section (or virtual wake cut) is defined over a greater distance than data is available. In sub figures c) and d) we show the wind field interpolated on the cross section. Outside of the measurement/scan area no data appears.

**9. [Reviewer #1]** Fig. 5b) and Fig. 6b): What is the meaning of the rotor like looking icons?
**[Authors]** The symbols mark curtailed or not operating turbines. We added the symbol in the captions of both figures.

**10. [Reviewer #1]** Fig. 8): . . . in the averaged lidar interval are marked by ? red? horizontal dotted lines?
**[Authors]** We added the word "red" in the captions of Figures 4 and 8.

**11. [Reviewer #1]** Fig. 9): Why is the power output only shown for the front row turbines? Are the other turbines producing less than rated power and the z-score is, hence, omitted for these turbines?
**[Authors]** The inner turbines in the wind farm experiences inner farm wake effects producing less power than in the front row. Since in this case the std. dev. of the front row turbines is small compared to the std. dev. of the whole wind farm we choose to focus on the front row turbines. Showing the power of turbines deep downstream in the wind farm would make it necessary to adjust the power scale to larger negative values. The effect intended to be shown here, the influence on the power of the first row in the cluster wake deficit, would be overlaid by the inner farm effects and appear less distinct. We replaced the symbol of the downstream turbines to a hexagon while the curtailed/not operational turbines in the front row remain with the old marker (Y).

**Nicolai Gayle Nygaard, Reviewer #2**

**1. [Reviewer #2]** P. 2, line 9-10: this makes it sound like optimization of wind farm layouts to reduce wake effects is new. Of course, the industry has been doing this for many years. I suggest slightly rephrasing to avoid this misinterpretation.

**[Authors]** We rephrased the sentence to "Optimized wind farm layouts on the basis of the prevailing wind rose and stability distribution to reduce wake effects are commonly used (e.g. Emeis, 2009; Turner et al., 2014; Schmidt and Stoevesandt, 2015)."

**2. [Reviewer #2]** P.2, line 15: I suggest adding a reference to the work by Volker on the Explicit Wake Parametrisation (EWP), for example Geosci. Model Dev., 8, 3715–3731, 2015

**[Authors]** We added the reference Volker et al. (2015) as suggested.

**3. [Reviewer #2]** P.3, line 13-14: maybe add that the wake and free flow regions are defined manually

**[Authors]** Added.

**4. [Reviewer #2]** P.4, line 16: clarify if the curtailment filter is based on a specific SCADA signal or derived from a combination of signals. This is of general interest to readers working with SCADA data. A reference to another paper describing similar filtering would also be sufficient

**[Authors]** We added the use of a SCADA status flag, a curtailment signal and the turbine's blade pitch angle to the manuscript.

**5. [Reviewer #2]** P.6, table 1: are the hub heights with reference to mean sea level?

**[Authors]** The values given are referenced to different height levels (MSL, LAT, "over water"). We added a hint on this fact in the caption and neglect the error here, since the difference between LAT and MSL is around 2 m in the North Sea.

**6. [Reviewer #2]** P.6, line just above table 2: insert "the" between "in" and "beam"

**[Authors]** Added.

**7. [Reviewer #2]** P.7: how does the finite acquisition time combined with the scan rate "smooth" the lidar measurements? Is this similar to a spatial averaging? Please comment in the text

**[Authors]** We added a sentence stating the spatial averaging perpendicular to the beam. "In both scenarios the laser beam is scanned over an angle of 2° per measurement leading to spatial averaging perpendicular to the line of sight direction."

**8. [Reviewer #2]** P.8, line 5: please add a description (and a reference if relevant) for the interpolation onto a regular grid. Also, it is not clear if the exclusion of grid points with less than 10 single scan contribution applies after interpolation or time averaging or both. Please clarify this

**[Authors]** We specified the interpolation method and better described the averaging procedure. Furthermore, we corrected the threshold for not contributing scans to accept a grid point to be valid. This threshold was chosen for scenarios A and B individually due to the different scan durations.

**9. [Reviewer #2]** Eq. (3): in Stull (1988) the bulk Richardson number is defined in terms of the virtual potential temperature. The authors use the potential temperature. Please specify why this approximation is appropriate and does not introduce bias in the classification of stability. Furthermore, I have seen other papers where the temperature in the denominator of the bulk Richardson number is not the surface temperature, but the air temperature at the measurement height. This cannot change unstable to stable conditions, but it can shift the stability parameter closer or further from neutral conditions. Finally, I wonder if the lidar measurements used in equation 3 include measurements in the cluster wakes. If this is the case, the wind speed in the denominator of equation 3 will be too small, thus biasing the stability parameter away from neutral conditions

**[Authors]** We specified and adapted the manuscript to clearly state the procedure of stability derivation. Furthermore we added a short appendix on how we derived the Richardson number

providing more detail. We corrected the temperature in the denominator of the bulk Richardson number to be the virtual potential temperature at the reference level following Emeis (2018).

The lidar measurements we used to derive the bulk Richardson number were recorded in cluster wakes, when present. We added a statement in the manuscript pointing this out. We do not see a general problem in this methodology, since our goal is to characterize the large-scale inflow of the wind farm, whether it is overlaid by a cluster wake or not. When the wake influences ambient stability, this is part of the inflow and needs to be accounted for.

**10. [Reviewer #2]** P. 9, line 13: which z is used to calculate the stability parameter. Is it z_TP? Please specify

**[Authors]** We specified it to be the height of the TP z_TP resulting in zeta = z_TP/L

**11. [Reviewer #2]** P.12, figure 5: are the non-operating/curtailed turbines the one marked with the symbols that are not circles? This is not clear from the caption. The same symbol is used in figures 9-11, but is not referred to in the captions. Were most turbines standing still or curtailed in the second flow case?

**[Authors]** We added the symbol to the Figure's caption. It marks the non-operating or curtailed turbines. In Figures 9 to 11 we decided just to show the power of the upstream turbines, since, due to inner farm effects, the scale of the z-score would reach further in the negative range when showing the power of the whole farm. This leads to less pronounced differences in the first row. Since the focus is here on the power of the upstream turbines, we changed the markers for the turbines behind the first row in the figure to hexagons just indicating turbine coordinates.

**12. [Reviewer #2]** p. 15, figure 8: I worry about the classification of weakly unstable conditions at the time of the 11 October 2018 SAR image. The stability is assessed entirely on a model. What is the uncertainty of this? Later in the same day, when meteorological measurements are available a large bias is seen between the mesoscale temperature and the measured temperature. This bias (if it also existed in the morning) could maybe change the stability classification from weakly stable to weakly unstable.

**[Authors]** The bias in the temperature data from the TP and NEWA has to be seen in the context of different reference heights. But you are right, the stability classification in the morning based on NEWA data alone is not very reliable. To support the NEWA data we obtained meteorological measurements from nacelles of some of the turbines in GTI. Even though these sensors are not calibrated and are most likely no first-class sensors the data gives more evidence on the stratification in the morning of 11.10.2018. Figure 1 displays the data together with the derived stability for four turbines. It confirms the unstable stratification in the morning and the transition to stable stratification after 12:00 UTC. All four temperature measurements are below sea surface temperature until approximately 12:00. We included temperature data and zeta of turbine GT58 in Figure 8 in the manuscript.

Additionally we looked into temperature profiles from radiosonde soundings as suggested in a short comment by Rémi Gandoin and Reviewer #1. Figure 2 displays soundings at the stations Bergen (04:00 UTC and 10:00 UTC) and Ekofisk (11:00 UTC, 300 km downstream of GTI in the North Sea, course approx. 317°) on 11.10.2018. The temperature profile at Bergen, 04:00 UTC, reveals decreased temperatures in the lower layer up to 300 m in the early morning due to heat radiating to space under a clear sky. This cooled air mass is transported by the south-eastern flow over the approximately 16 °C warm North Sea leading to a shallow unstable boundary layer up to approx. 200 m to 300 m height where a strong inversion with increasing temperatures starts and blocks convection to reach up higher. This shallow unstable layer still appears in the sounding at Ekofisk, 11:00 UTC, but disappears over land during the day due to solar warming of the ground, see temp Bergen, 10:00 UTC.

We added a short statement on the radiosonde soundings and the sources for sounding data in the paper without showing the temperature profile.

In conclusion we stick to our statement of a shallow (weakly) unstable stratified boundary layer in the morning of 11.10.2018. In the discussion we added a part pointing out the necessity of a careful characterization of atmospheric stability and consideration of boundary layer height.

[Figure]

*Figure 1: Meteorological measurements and stability classification on 11.10.2018. Measurements taken on turbines GT51, GT58, GT59 and GT64 in Global Tech I. Top to bottom: relative humidity, temperature, pressure (only single measurement available), stability parameter zeta (note erroneous values for bulk Richardson numbers above critical value of 0.2), air speed on the nacelle and the bulk Richardson number.*

[Figure]

*Figure 2: Temperature profiles obtained from radiosonde soundings at Bergen (station nr. 10238) and Ekofisk (station nr. 1400) at 11 October 2018. Dry adiabatic curves (slope -100 m/°C, orange dotted lines and dash dotted line) and the sea surface temperature of approximately 16 °C (vertical dashed line) are drawn. [Data: www.meteociel.fr]*

**13. [Reviewer #2]** p. 16, line 30: can you comment on the expected AEP impact of the OSS platforms? I would expect it to be small, since the platforms are fairly low compared with the turbines and have a smaller cross-sectional area.
**[Authors]** We did not analyse the impact of platform wakes on the power of neighbouring or distant wind turbines. Therefore we could not comment neither on the impact on single turbines nor on the AEP of a wind farm. We agree with the assumption of a fairly low effect compared to wind turbine wakes due to the lower height and the smaller cross section. We added a short comment on this matter in the end of section 4.2.

**14. [Reviewer #2]** p. 17, line 3: at rated power OR above rated speed (as opposed to at rated speed)
**[Authors]** We changed the phrase to "above rated speed".

**15. [Reviewer #2]** p. 17, line 9: if the average wind speed is smaller the wind speed deficit should be larger not smaller due to the increase of the thrust coefficient at lower wind speeds. Or am I missing something?
**[Authors]** We clarified the statement in the manuscript stating a similar relative wake deficit and a higher absolute deficit for higher wind speeds. The turbine's thrust coefficient should be more or less constant in the partial load range leading to similar relative wake deficits. The DolWin2 cluster operated in partial load at the regarded times (c.f. www.energy-charts.de).

**16. [Reviewer #2]** p. 18, line 7: when saying that SAR mostly supports the lidar wake measurements, can you be more specific?
**[Authors]** We did not perform a systematic analysis of all available lidar measurements and corresponding SAR data, yet. This is planned for future work. Aside cases where SAR data and lidar agrees well, in some cases we found cluster wake like structures in the lidar while no conclusive signature of cluster wakes were evident in the SAR data. This complicates the interpretation of the situation due to the limited range of the lidar measurement (as discussed in the manuscript). We rephrased the manuscript to "In some of the cases with available large-area SAR wind data these alternative measurements supported the lidar cluster wake measurements.".

**17. [Reviewer #2]** p. 21, line 14: I suggest this phrasing: Wind turbines are sensitive to the wind conditions over a wide range of heights defined by the swept rotor area.
**[Authors]** We changed the sentence to the proposed wording.

**18. [Reviewer #2]** p. 22, line 9: is the wind speed deficit region truly decreasing in width, or is it the region of a certain colour in the heat map that is shrinking? The increase of wake width is typically coupled with the decrease of the peak deficit.
**[Authors]** The colours in Figure 9 represent the SAR wind speed. The regions with the highest deficits (dark blue colours) extend downstream from the farms/clusters and become narrower with increasing distance. These regions do not necessarily represent the full wake but the inner region with the highest deficit. The outer wake regions seem so smear with the surrounding flow. We clarified our discussion on this point.

**19. [Reviewer #2]** p. 23, line 26: "then" should be "than"
**[Authors]** corrected

[revised manuscript text omitted]

---

## Author Response (AR2)

**Cluster wakes impact on a far distant offshore wind farm's power**

Authors: JörgeSchneemann, Andreas Rott, Martin Dörenkämper, Gerald Steinfeld, Martin Kühn

DOI: 10.5194/wes-2019-39

**Author response to editor comment**

Thank you for your comment. We included the two references in the manuscript. Furthermore we corrected a few typos and shifted labels in Figure 1b to improve clarity.

All changes to the text are highlighted in the LaTeX-Diff document below.

[revised manuscript text omitted]